# Multi-scale time series prediction model based on deep learning and its application

**Zhifei Yang**[1,2]*, **Jia Zhang**[1,2], **Zeyang Li**[1,2]

**1** School of Electronic and Information Engineering, Lanzhou Jiao tong University, Lanzhou, China,
**2** Gansu Urban Traffic Big Data Application Industry Technology Center, Lanzhou Jiao tong University, Lanzhou, China

* 916408624@qq.com

**Data availability statement:** The minimal dataset necessary to replicate all study findings, including underlying values, analytical data points, and associated metadata, is publicly

## Abstract

Time series prediction is a widely used key technology, and traffic flow prediction is its typical application scenario. Traditional time series prediction models such as LSTM (Long Short- Term Memory) and CNN (Convolution Neural Network)-based models have limitations in dealing with complex nonlinear time dependencies and are difficult to capture the complex characteristics of traffic flow data. In addition, traditional methods usually rely on manually designed attention mechanisms and are difficult to adaptively focus on key features. To improve the accuracy of time series prediction, the paper proposes a multiscale convolutional attention long short-term memory model (MSCALSTM), which combines a multiscale convolutional neural network (MSCNN), a multiscale convolutional block attention module (MSCBAM) and LSTM. MSCNN can effectively capture multiscale dynamic patterns in time series data, MSCBAM can adaptively focus on key features, and LSTM is good at modeling complex time dependencies. The MSCALSTM model makes full use of the advantages of the above technologies and greatly improves the accuracy and robustness of time series prediction. Extensive experiments are conducted on a dataset from the California Performance Measurement System (PEMS), and the results show that the proposed MSCALSTM model outperforms the state-of-the-art models. Experiments in the Energy domain show that our model also has strong generalization properties in other time series forecasting domains.

## Introduction

Accelerated urbanization has made traffic congestion a major challenge for large cities. While significant resources are devoted to mitigating congestion and its associated problems, most solutions are costly, difficult to implement, or both. In contrast, accurate traffic flow prediction offers a cost-effective and practical alternative. This approach utilizes historical traffic data to forecast future conditions, such as flow and speed. By analyzing and processing this data, predictions of future traffic states enable traffic management departments to implement flexible control strategies, thereby reducing large-scale congestion and improving travel comfort.

available without restriction on GitHub:
https://github.com/lulu-dududu/code.git.

**Funding:** This work was supported by the Humanities and Social Sciences Research Project of the Ministry of Education (20YJCZH212 to YZF), and the Experimental Teaching Reform Project of Lanzhou Jiao tong University (2024013 to YZF).

**Competing interests:** The authors have declared that no competing interests exist.

The development of traffic flow prediction methods has evolved from traditional statistical approaches to machine learning and eventually deep learning. Initially, statistical methods like Auto Regressive Integrated Moving Average (ARIMA) [1] and Kalman filtering [2] were employed, using historical data analysis for prediction; while simple and intuitive, their effectiveness is often limited with complex, non-stationary traffic patterns. Subsequently, machine learning methods such as Support Vector Machine (SVM) [3] and K-Nearest Neighbors (KNN) [4] emerged, capable of capturing nonlinear relationships and hidden patterns through model training. However, these traditional machine learning approaches frequently suffer from overfitting due to the inherent nonlinearity of traffic flow data, which negatively impacts their prediction performance. In recent years, deep learning-based traffic flow prediction methods have emerged as a research hotspot. Convolutional Neural Networks (CNN) [5,6] are widely used for capturing crowd flow characteristics via grid analysis, demonstrating superior feature extraction and information mining capabilities compared to other deep learning approaches.Recurrent Neural Networks (RNN) [7,8] inherently suffer from long-term dependency issues, which are effectively addressed by Long Short-Term Memory networks (LSTM) [9]. Have become the most prevalent method for time series forecasting due to their ability to comprehensively model nonlinear characteristics. This is evidenced by successful LSTM applications in diverse domains, such as cryptocurrency price classification [10], short-term subway passenger flow prediction [11], and emotion EEG recognition with enhanced feature extraction [12].

However, standalone LSTM models exhibit limitations in achieving high prediction accuracy and fully capturing complex data features for time series tasks. This has prompted researchers to integrate LSTM with complementary methods, forming composite models to enhance performance [13–15]. A particularly successful approach combines the spatiotemporal feature extraction strengths of CNN and LSTM; for instance, composite CNN-LSTM architectures have demonstrated superior accuracy over single models in diverse applications like capturing nonlinear relationships [16], power consumption forecasting [15], water quality prediction [17], and gold price fluctuation analysis [18]. Concurrently, attention mechanisms have been leveraged to boost model precision by enabling focused processing of critical input subsets, implemented in structures ranging from encoder-decoder based networks for recursive models [19] to lightweight modules like the squeeze-excitation network (SeNet) applied in convolutional networks such as ResNet [20,21].The convolutional block attention module (CBAM) [22] is an improvement on SeNet, which considers global average and maximum pooling of channel and spatial attention modules at the same time. Nauta et al. [23] considered attention-based dilated deep separable temporal convolutional networks (AD-DSTCNs) and demonstrated that the attention mechanism can be effectively applied to time series prediction of (Dynamic Deep Spatial-Temporal Convolutional Network) DDSTCN. Spatial-Temporal Graph Convolutional Network (STGCN) [1] and Diffusion Convolutional Recurrent Neural Network(DCRNN) [24] use the distance between actual road network nodes to represent the spatial correlation between road segments. Ma et al. proposed Spatial-Temporal Adaptive Graph Convolutional Network(STAGCN) [25] to automatically capture the spatiotemporal relationship in traffic sequences and adaptively model the road network spatial topology graph. Temporal-Fusion Graph Convolutional Attention Mechanism(TFM-GCAM) [26] is based on the traffic flow matrix and consists of graph convolution and attention mechanism. It captures the traffic characteristics of nodes more clearly and reduces the computational cost.

Although the above models can capture spatial dependencies and better model structured time series data using graph convolution compared to basic single models, they still lack the ability to model multi-scale spatial dynamic patterns, and the attention mechanism

used needs to be manually designed and lacks adaptability. Transformer-based methods [27] use self-attention mechanisms to effectively model long-term spatiotemporal dependencies and achieve good results, but they lack the ability to model spatial features and multi-scale temporal dynamic patterns. Recent spatiotemporal modeling frameworks leveraging dynamic graph structures have shown significant advantages in extracting feature correlations. Notably, He et al. [28] developed the Dual-Correlation Dynamic Graph Convolutional Network (DC-DGCN), which employs multi-objective optimization to dynamically update graph structures and integrates bidirectional LSTM for real-time physical-digital space interaction, effectively capturing multiscale dynamic degradation patterns through coordinated learning. Complementary hybrid approaches also demonstrate strong performance: a novel architecture combining machine learning with wavelet transform [29] excels at capturing long-term dependencies across power datasets, while a hybrid model fusing wavelet-transformed data with an artificial neural network (integrating long/short-term networks) [30] achieved state-of-the-art results on greenhouse gas monitoring data from Russia's Bely Island.Multi-scale feature extraction methods have also advanced significantly in industrial applications; for instance, He et al.'s RTSMFFDE-HKRR method [31] achieves high-precision bearing fault diagnosis in noisy environments through multi-scale entropy and regression techniques, focusing on local feature decoupling and signal stability. In contrast, our model employs parallel multi-scale convolutional kernels to coordinate the extraction of road-level micro-features and regional-level macro-features, dynamically weighting them via a spatiotemporal attention mechanism. This design avoids information loss inherent in traditional entropy-based multi-scale methods (e.g., RTSMFFDE's coarse-graining) and leverages convolution's translation invariance to directly model traffic flow spatial propagation.

Nevertheless, despite the high accuracy offered by existing deep learning methods, their substantial computational burden and training complexity remain challenges. To address this and enhance time series prediction accuracy and robustness, this paper proposes MSCALSTM—an attention-based composite model integrating Multi-Scale CNN (MSCNN) for extracting multi-scale dynamic patterns, Multi-Scale Convolutional Block Attention Module (MSCBAM) for adaptively focusing on key predictive features, and LSTM for modeling complex temporal dependencies. The key contributions are:

(1) We proposes a time series prediction method that combines multi-scale CNN, multi-scale CBAM and LSTM to improve the prediction accuracy; (2) We introduces a multi-branch convolutional layer in CBAM for improvement. Using convolutions of different scales can enable the module to better model the spatial relationship between features; (3) Experimental results show that compared with existing methods, MSCALSTM outperforms the state-of-the-art methods on four public datasets in the transportation field (PeMSD3, PeMSD4, PeMSD7, and PeMSD8) and on a dataset in the Energy field. The rest of this paper is organized as follows: Section 2 reviews the related work based on traffic flow prediction; Section 3 systematically describes the overall structure and specific methods of the MSCALSTM model; Section 4 introduces four commonly used traffic flow datasets and one energy dataset, and gives the evaluation metrics; Section 5 discusses the experimental results; finally, Section 6 concludes this paper.

## Related work

### Convolutional neural network

CNN (Convolutional Neural Network) is widely used in image processing and computer vision, and has also been shown to be suitable for the study of traffic flow prediction problems [32]. The typical CNN architecture consists of convolutional layers, pooling layers, dropout

layers, and fully connected layers [33]. CNN uses convolutional kernels to capture relevant features from the input data, which are then processed through pooling layers and fully connected layers to produce the final output. GCN (Graph Convolutional Network) is a typical example of a CNN model. Zhao et al. used GCN to capture the spatial dependencies of roadmaps and constructed a TGCN (Temporal Graph Convolutional Network) model [34]. Peng et al. [35] proposed a spatiotemporal correlation dynamic graph recurrent CNN to predict urban traffic passenger flow. However, CNN has limitations in capturing the temporal dependencies of time series data. Therefore, it is necessary to integrate recurrent neural network (RNN) technology and combine CNN with long short-term memory (LSTM) networks to improve the performance of prediction tasks [36]. The structural diagram of CNN is shown in Fig 1:

## Long short memory network

LSTM (Long Short-Term Memory) is a deep learning structure and a variant of RNN (Recurrent Neural Network). RNN can be applied to the prediction field of time series data [37]. Since RNN has the problem of gradient vanishing, the gating mechanism of LSTM can solve this problem [9], and LSTM has been applied to the field of traffic flow prediction. Ma et al. [38] used the LSTM model to construct a traffic speed prediction model. The experimental results show that the prediction accuracy of the LSTM network is better than most statistical learning methods. Zhao et al. [39] proposed a traffic flow prediction model based on LSTM, which takes into account the spatiotemporal correlation of the traffic system. The experimental results show that the model has better prediction performance than other mainstream prediction models. Tian et al. [40] proposed a multi-scale smoothing method to fill the missing values in traffic flow data, and based on this, established an LSTM model to predict traffic flow. Wang et al. [41] developed an LSTM encoder-decoder model combined with the attention mechanism of time series prediction. The model covers periodic patterns and recent time models. The results show that the model has high effectiveness and reliability in long-term time series prediction. Zhang et al. [42] proposed a short-term traffic prediction model that combines graph convolution operations and residual LSTM structures. The prediction effect of this model is better than six baseline models through evaluation on the traffic speed

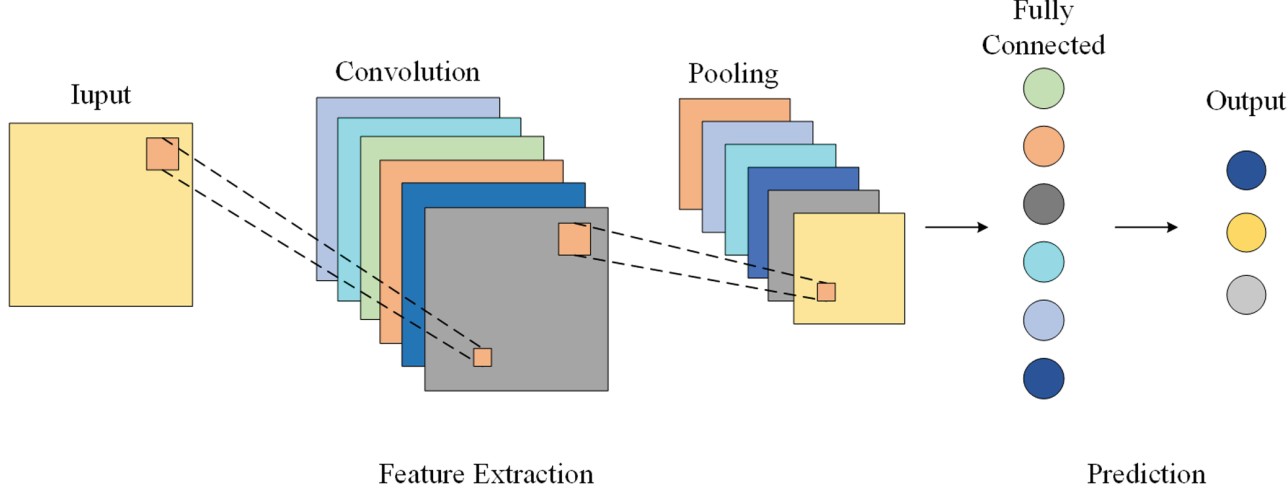

**Fig 1. CNN structure.**

dataset. Xia et al. [43] proposed a distributed LSTM weighted model that combines time window and normal distribution to improve the traffic flow prediction ability. The experimental results show that the model has improved the prediction accuracy.

## Attention mechanism

With the remarkable success of attention mechanism in many fields, researchers have also begun to apply it to traffic flow prediction. In the field of traffic flow prediction, many models have adopted attention mechanism in spatiotemporal dimension. Guo et al. [44] proposed an attention-based spatiotemporal graph convolutional network ASTGCN to more effectively capture the spatiotemporal relationship between nodes. Zhang et al. [45] proposed a spatiotemporal convolutional graph attention network (ST-CGA) to better extract the dependencies between global regions. Ali et al. [46] designed an attention-based network to capture the dynamic spatiotemporal correlation of traffic flow, thereby improving the prediction effect. Wang et al. [47] proposed an attention-based spatiotemporal graph attention network (ASTGAT) to solve the problems of network degradation and over-smoothing and enhance the ability to capture the dynamic correlation of traffic flow prediction. Qiu et al. [48] proposed an event-aware graph attention fusion network to effectively capture the spatiotemporal characteristics of traffic networks.

CBAM (Convolutional Block Attention Module) is a simple and efficient convolutional neural network attention module. It is able to generate attention feature maps in the two-dimensional domain of channel and feature space and adaptively adjust the original features. Since CBAM is a general end-to-end module, it can be easily integrated into the convolutional layer and jointly trained with the basic convolutional layer [49]. CBAM [22] enhances feature connectivity across both channel and spatial dimensions through sequential application of channel and spatial attention modules. Wang et al. [50] proposed a generative adversarial network (GAN) that combines high-order degradation models and CBAM, aiming to generate low-resolution remote sensing images, significantly improving image quality. This integration strategy effectively reduces noise interference and achieves significant improvements in texture and feature representation. In addition, Wang et al. [51] also proposed a model called MTCNet that combines multi-scale transformers with CBAM to improve the detection quality of different remote sensing image change detection tasks.

CBAM has become one of the most widely used attention mechanisms due to its high efficiency and lightweight. This mechanism takes into account both the channel dimension and the spatial dimension, effectively highlighting important features and compressing redundant features. The channel attention and spatial attention submodules are the core components of CBAM. The basic structure of each submodule is a multilayer perceptron (MLP), which includes an input layer, a hidden layer, and an output layer. Fig 2 shows the detailed structure of the network.

The channel attention module compresses the feature map in the spatial dimension (width and height), combining average pooling and maximum pooling. After the pooled vector is processed by a multi-layer perceptron (MLP) with shared weights, the element-by-element product of the output vector is added. Finally, the channel attention feature vector is obtained by the sigmoid activation function. The specific process is shown in formula (1):

$$M_C(F) = \sigma(MLP(AvgPool(F)) + MLP(MaxPool(F))) \tag{1}$$

where $M_C$ is the channel attention vector, indicating the importance of the channel, $F$ is the feature map, AvgPool represents the average pooling, MaxPool represents the maximum pooling and $\sigma$ represents the activation function of the sigmoid.

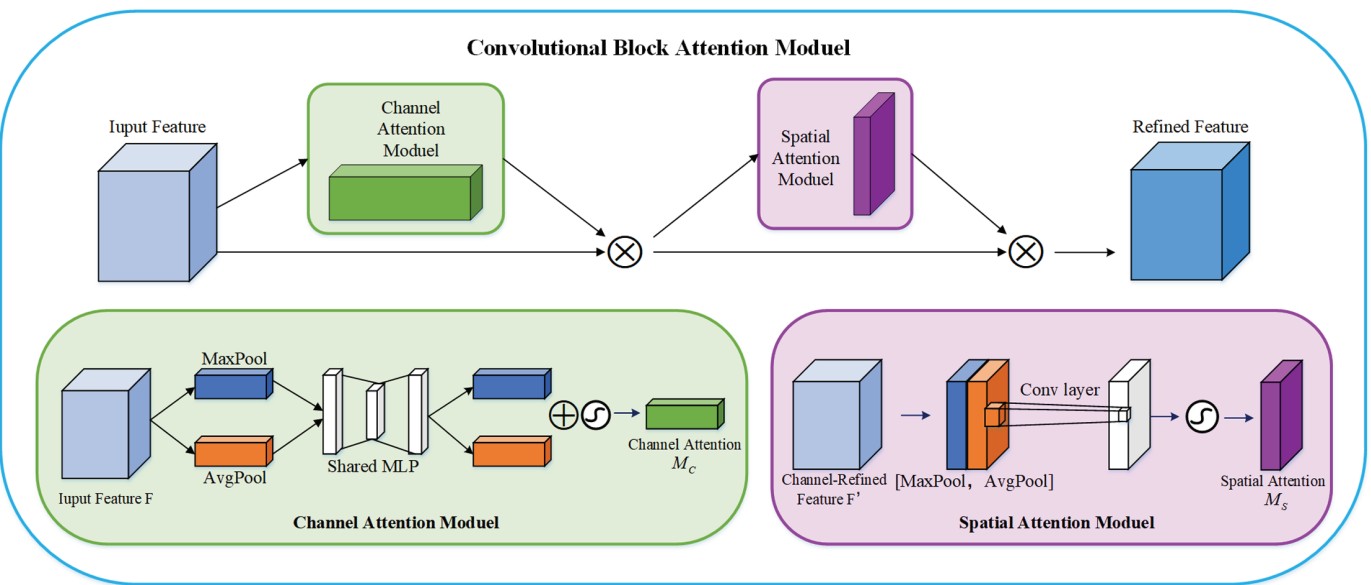

**Fig 2. CBAM structure.**

Spatial attention focuses on important spatial locations and performs average pooling and maximum pooling in the channel dimension. This module concatenates the two pooling matrices and then mixes them through a convolutional layer to reduce the number of channels. Finally, the result is processed by a sigmoid activation function to generate a spatial attention matrix. The specific process is shown in formula (2):

$$M_S(F) = \sigma\left(Conv\left([AvgPool(F); MaxPool(F)]\right)\right) \tag{2}$$

Among them, Conv is the convolution layer and $M_S$ is the spatial attention vector.

The CBAM attention mechanism effectively combines the channel attention module with the spatial attention module, which has been proven to achieve superior results. First, the channel attention module takes the feature map F as input and outputs the attention matrix $M_C(F)$. By multiplying these two components, a new feature map $F'$ is generated. Next, the output $F'$ of the previous stage is input into the spatial attention module, and the process is repeated to obtain the final feature map $F''$. Its mathematical expression is shown in formula (3):

$$\begin{cases} F' = M_C(F) \otimes F \\ F'' = M_S(F') \otimes F \end{cases} \tag{3}$$

Among them, $M_C$ is the channel attention vector, $\otimes$ is the element-wise multiplication method, and $M_S$ is the spatial attention vector.

## Method overview

This section will elaborate on the proposed method MSCALSTM, which can be divided into three main modules. The first module is the multi-scale convolution (MSCNN), which can extract multi-scale features, increase the receptive field, improve prediction accuracy and

enhance generalization ability through multi-branch convolution. The second module is the multi-scale convolution block attention mechanism (MSCBAM), which can enhance the key feature representation and adaptively focus on important areas, thereby improving the model's prediction performance and interpretability for time series data. The core of the discussion on enhancing interpretability of MSCBAM is that the spatial attention map generated by multi-scale CBAM can intuitively reflect the model's focus on key areas in traffic flow data. Specifically, in the traffic flow prediction task, MSCBAM uses multi-scale convolution ($3 \times 3$ and $5 \times 5$ convolution kernels) to capture spatial dependencies of different ranges, and adaptively assigns weights through the attention mechanism to visualize the model's attention to important intersections or road sections. For example, the attention map can clearly show the model's dependence on surrounding intersections or main roads when predicting traffic flow in a certain area, thereby revealing the key spatiotemporal dependencies in traffic flow. This design enhances the interpretability of the model. The third module is the LSTM and fully connected layer, which can capture the complex dependencies in the data, perform dimensionality transformation and feature combination, and generate the final prediction results. In this way, the proposed MSCALSTM can effectively focus on key features, enhance nonlinear modeling capabilities, and better adapt to the complexity of data. Specifically, The method combining multi-scale CNN, multi-scale CBAM and LSTM performs well in time series forecasting because it can effectively extract multi-scale features and capture short-term and long-term trends. Multi-scale CNN provides a combination of local and global information, while the attention mechanism of multi-scale CBAM dynamically adjusts feature weights, enhances the focus on important features, and better models the spatial relationship between features. LSTM is good at capturing temporal dependencies and can handle complex dynamic changes. The combination of the three makes the model more expressive in the fusion of spatial and temporal features, thereby significantly improving the prediction accuracy. The overall framework of MSCALSTM is shown in Fig 3. Among them, $\oplus$ in MSCBAM represents concatenation and $\otimes$ represents residual connection. $\otimes$ in LSTM represents multiplication, and $\sigma$ represents sigmoid.

## Multi-scale convolution

First, the data is usually used as the input of the model in the form of (B, T, C), where B represents the batch size, T represents the time step, and C represents the number of features. In order to expand the input image size and better realize feature extraction, this paper introduces batches into the operation for the first time, and adjusts the data format to $X \in R^{C \times B \times T}$ through dimensionality transformation. The input data is first processed by

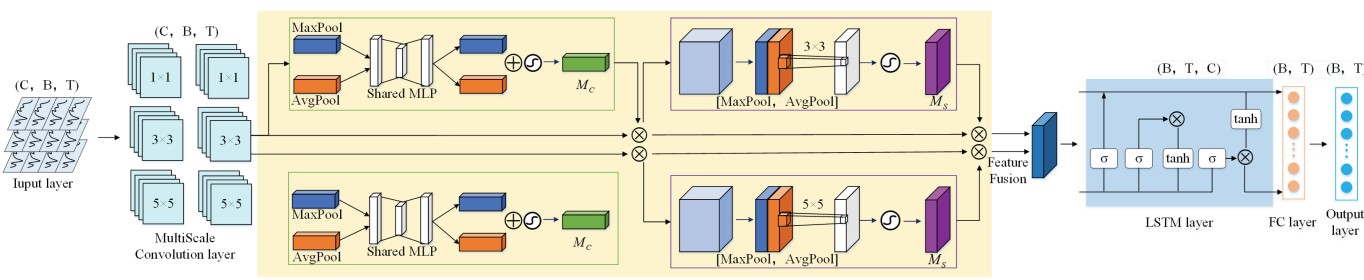

**Fig 3. MSCALSTM.**

MSCNN for multi-scale convolution, and then the concatenated result is reduced in dimension. The dimension after dimensionality reduction is still (C, B, T). Specifically, three convolution kernels of size 1×1, 3×3 and 5×5 are set. Multi-scale convolution is shown in formula (4):

$$F = Concat(f_{k_1}(X), f_{k_3}(X), f_{k_5}(X)) \tag{4}$$

where $f_{ki}$ represents the convolution operation with a convolution kernel size of $k_i$, and Concat represents the concatenation operation.

Specifically, using convolution kernels of different sizes can capture features of different scales, enhance the richness of feature representation, effectively expand the receptive field, and enable the model to capture longer-range dependencies, thereby improving prediction accuracy. $1 \times 1$ convolution focuses on the relationship between channels, $3 \times 3$ convolution is suitable for capturing local details, and $5 \times 5$ convolution can extract more extensive contextual information.

## Multi-scale CBAM

The improvement of introducing multi-branch convolutional layers in CBAM enables the model to extract features of traffic flow data at different scales simultaneously, thereby more effectively modeling the spatial relationship between features. This design enhances the focus on important features related to traffic flow prediction, improves the flexibility and expressiveness of the model under complex traffic patterns, and significantly improves the accuracy of prediction.

Expanding the dimension enables the data to be better processed by the CBAM attention module, and the expanded dimension is in the form of (1, C, B, T). Then, the dimensionalized data is input into the MSCBAM attention module. Specifically, The multi-scale channel attention mechanism (MSCBAM) used in this paper improves on the traditional spatial attention module and uses convolution kernels of two different scales, $3 \times 3$ and $5 \times 5$, to extract spatial features. This multi-scale convolution operation enhances the model's ability to model the spatial relationship between features, thereby improving performance. MSCBAM weights the extracted features through the spatial attention mechanism. Similar to the traditional CBAM, MSCBAM calculates the importance of each spatial position. However, its uniqueness lies in the generation of spatial attention maps of different scales through $3 \times 3$ and $5 \times 5$ convolution layers. This design enables the model to effectively integrate local and global information, thereby more accurately identifying the most critical areas for traffic flow prediction.

The result after multi-scale convolution is reduced in dimension through $1 \times 1$ convolution, and then the result after dimension reduction is passed through multi-scale CBAM to effectively enhance the key feature representation and adaptively focus on important areas, thereby improving the prediction performance and interpretability of the model for time series data. The MSCBAM attention operation can be expressed by equations (5).

$$\begin{cases} F_C = M_C(F) \otimes F \\ F_{MS} = M_{MS}(F_C) \otimes F_C \end{cases} \tag{5}$$

In the above formula, $F$ uses the dimension reduction result as the input feature map of the MSCBAM attention module. $\otimes$ represents the multiplication between elements, $M_C$ is the channel attention extraction operation, and $M_{MS}$ is the multi-scale spatial dimension extraction operation.

The CAM attention calculation formula is shown in (6), where $\sigma$ is the sigmoid function.

$$M_C(F) = \sigma(MLP(AvgPool(F)) + MLP(MaxPool(F))) \tag{6}$$

The formula of the MSSAM attention module is shown in (7), where $M'_{S_1}(F')$ represents the MSSAM result using a $3 \times 3$ convolution kernel, and $M'_{S_2}(F')$ represents the MSSAM result using a $5 \times 5$ convolution kernel. $M'_S(F')$ represents the result after concatenating and reducing the dimension of the MSSAM results of the two convolution kernels.

$$\begin{cases} F' = M_C(F) \\ M'_{S_1}(F') = \sigma(f^{3\times3}([AvgPool(F); MaxPool(F)])) \\ M'_{S_2}(F') = \sigma(f^{5\times5}([AvgPool(F); MaxPool(F)])) \\ M'_S(F') = Conv^{1\times1}(Concat(M'_{S_1}(F'), M'_{S_2}(F'))) \end{cases} \tag{7}$$

Among them, Concat represents the concatenation operation, $Conv^{1\times1}$ represents the $1 \times 1$ convolution, and $\sigma$ represents the sigmoid function.

Finally, the dimension of the reduced result is converted to the original state (B, T, C) through the reshaping operation so that it can be better processed by the subsequent LSTM.

## LSTM

The long short-term memory (LSTM) method consists of three gates: a forget gate, an input gate, and an output gate [9]. The forget gate uses the Sigmoid function to selectively filter the memory information of the previous moment and the new input information. When the value of the gate is 1, all information is retained; when the value is 0, all information is discarded. The design of this gate effectively alleviates the common gradient vanishing problem in the recurrent neural network (RNN) model. LSTM combines multiple gating mechanisms, such as the forget gate, to selectively retain important information from previous time steps and discard irrelevant information. This feature is crucial to LSTM's ability to retain long-term dependencies in sequence data. The LSTM structure diagram is shown in Fig 4.

The result $M'_S(F')$ after MSCBAM processing is used as the input of LSTM. The formula for processing in LSTM is shown in formula (8):

$$\begin{cases} x_t = M'_S(F') \\ f_t = \sigma(W_f \cdot [h_{t-1}, x_t] + b_f) \\ i_t = \sigma(W_i \cdot [h_{t-1}, x_t] + b_i) \\ \widetilde{C}_t = \tanh(W_C \cdot [h_{t-1}, x_t] + b_C) \\ C_t = f_t * c_{t-1} + i_t * \widetilde{C}_t \\ o_t = \sigma(W_O \cdot [h_{t-1}, x_t] + b_o) \\ h_t = o_t * \tanh(C_t) \end{cases} \tag{8}$$

Among them, $\sigma$ and tanh represent the sigmoid function and tanh function respectively; $x_t$ is the input of the LSTM unit at time t. It is worth noting that $x_t$ represents the feature $M'_S(F')$ extracted by the MSCBAM proposed in this paper; $W_f, W_i, W_c, W_o$ represent the weight matrices of the forget gate, input gate, input node and output gate respectively; $b_f, b_i, b_c, b_o$ represent the bias vector of each gate respectively; $h_{t-1}$ and $h_t$ represent the hidden states of the LSTM unit at time $t-1$ and time $t$; $f_t, i_t, c_t$ and $o_t$ represent the outputs of the forget gate, input gate, input node and output gate respectively; $\widetilde{C}_t$ is the candidate unit state at time t; $C_t$ and $C_{t-1}$ represent the unit states at time $t$ and $t-1$.

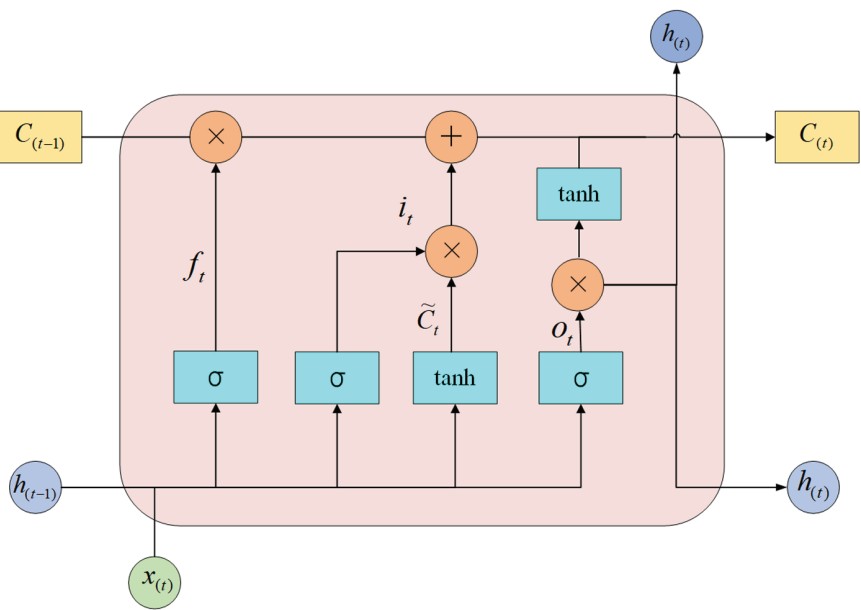

**Fig 4. LSTM.**

Finally, a fully connected layer is used to transform the dimension of the feature vector output by LSTM and map it to the dimension of the prediction target. At the same time, the output features are further extracted and combined to enhance the nonlinear modeling capability and output the final prediction result with the dimension (B, T).

## Experimental analysis

### Datasets

The proposed model is validated on four highway datasets and energy datasets in California, namely PeMSD3, PeMSD4, PeMSD7, PeMSD8 and Energy.

> PeMSD3: This dataset contains traffic flow data from 358 traffic collection nodes from September 1, 2018 to November 30, 2018.
>
> PeMSD4: It refers to the traffic data of the San Francisco Bay Area, which contains 3848 detectors on 29 roads. The time span of this dataset is from January to February 2018. This paper selects the first 50 days of the dataset as the training set, and the remaining data as the test set.
>
> PeMSD7: The traffic speed dataset is collected by the California Department of Transportation in the seventh district of California through 228 road traffic sensors, and the collected data samples are aggregated every 5 min. The dataset records the vehicle speed of the seventh district of California from May 1, 2012 to June 30, 2012.
>
> PeMSD8: It is the traffic data released in San Bernardino from July to August 2016, which contains 1979 detectors on 8 roads. This paper selects the first 50 days of data as the training set, and the last 12 days of data as the test set.
>
> Energy: It includes the monthly natural gas production data of a gas field in southwest China from 1992 to 2021.

## Baseline model

This article compares the following models with the proposed MSCALSTM model:

**HA** [52]**:** This model uses the weighted average of historical speed data to predict future speeds.

**ARIMA** [53]**:** This model is a classic time series prediction model.

**LSTM:** This model maintains data validity features through a gating mechanism.

**STGCN** [1]**:** This model is a convolutional structure for traffic prediction tasks based on spatiotemporal GCN.

**DCRNN** [24]**:** This model is a classic GNN-based spatiotemporal series prediction method that uses diffuse convolutional RNN to handle complex spatial dependencies and nonlinear temporal correlations in road networks.

**ASTGCN** [44]**:** This model is an evolution of STGCN, which introduces the spatiotemporal attention mechanism of STCGCN.

**STSGCN** [54]**:** This model is a spatiotemporal synchronous graph convolutional network that learns local spatiotemporal correlations through a spatiotemporal synchronous modeling mechanism.

**AGCRN** [55]**:** This model is an adaptive GCN that learns spatiotemporal dependencies from spatiotemporal data and can perform graph convolution without a predefined spatial graph.

**STG-NCDE** [56]**:** This model uses two differential neural control equations to handle the temporal and spatial dimensions of traffic prediction respectively.

**MAGRN** [57]**:** This model is based on a multi-scale graph convolutional network recursive feature extraction framework with multi-scale attention and a dual attention mechanism for traffic flow prediction.

**TARGCN** [58]**:** This model exploits the dynamic spatial correlation between traffic nodes and the temporal dependency between time slices for prediction.

## Evaluation metric

Evaluation metrics are an important part of determining model performance. In order to evaluate the prediction effect of MSCALSTM, this paper uses MAE (mean absolute error), RMSE (root mean square error), MAPE (mean absolute percentage error), AIC (Akaike's Information Criteria) and BIC (Bayesian information criterion) as the evaluation criteria of the evaluation method.

MAE is a metric equal to the mean absolute value of the difference between the truth $y_i$ and the predicted value $\widehat{y}_i$, and its definition is shown in formula (9):

$$MAE = \frac{1}{N}\sum_{i=1}^{n}|y_i - \widehat{y}_i| \tag{9}$$

RMSE is the standard deviation of the prediction error presented in the data set around the prediction result, and its definition is shown in formula (10):

$$RMSE = \sqrt{\frac{1}{N}\sum_{i=1}^{N}(y_i - \widehat{y}_i)^2} \tag{10}$$

Among them, $\widehat{y}_i$ is the predicted value and $y_i$ is the truth.

MAPE measures the size of the error in percentage terms and is defined as shown in formula (11):

$$MAPE = \frac{1}{N} \sum_{i=1}^{N} \frac{|y_i - \widehat{y_i}|}{y_i} * 100\%$$  (11)

Among them, $\widehat{y_i}$ is the predicted value and $y_i$ is the truth.

AIC is a statistical metric used to quantify the trade-off between model complexity and goodness of fit. The specific formula for AIC is shown in (12). BIC is an alternative model selection criterion that introduces a stronger penalty for model complexity. The specific formula for BIC is shown in (13).

$$AIC = 2k - 2\log(L^\wedge)$$  (12)

$$BIC = k\log(n) - 2\log(L^\wedge)$$  (13)

where $k$ is the number of estimated parameters, $n$ is the number of recorded measurements, and $L$ is the value of the likelihood. Both AIC and BIC are designed to decrease the model complexity by selecting the model with the lowest probability distribution.

## Implement details

All experiments are conducted in PyTorch with the NVIDIA GeForce 4090 GPU and 48G memory. The dataset is divided with a ratio of 6:2:2 into training sets, validation sets, and test sets. We train for 200 epochs using the Adam optimizer, with a batch size of 64 on all datasets. We adopt the Adam optimizer with an initial learning rate of 0.003 to train the model. The learning rate will be decayed at the steps of 15, 40, 70, 105, 145 and 190 with a decay rate of 0.3. Moreover, the model parameters with the lowest loss on the verification set are saved as the best parameters and tested on the test set. To verify the applicability of the model on large-scale data sets, this paper further tests the computational efficiency of MSCALSTM under different data scales. Since traffic flow prediction and energy prediction often involve massive data, in order to better simulate large-scale data scenarios, we expand the original data set through time series generation algorithm and spatial dimension splicing respectively.

Time series generation algorithm: In order to expand the traffic flow data, we use the time series generation algorithm to expand the original data to generate large-scale simulation data. Similarly, we also use this method to expand the energy data set.

Spatial expansion: The detector data of PeMSD3 (358 nodes) and PeMSD7 (228 nodes) are horizontally spliced to construct a 586-node virtual road network (PeMSD3+7) to simulate the complexity of the city-level road network.

In order to verify the performance and scalability of the model in large-scale data scenarios, this paper designed a series of experiments through data expansion and optimization strategies. Through the time series generation algorithm, PeMSD4 (3848 nodes, 50 days of training data) and PeMSD8 (1979 nodes, 62 days of complete time series) were expanded to double the duration, and long sequence synthetic datasets containing 100 days (PeMSD4Large) and 124 days (PeMSD8Large) were constructed. As shown in Table 1, the single CPU training time of the expanded PeMSD4Large increased from 2.8 hours to 5.9 hours, and the memory usage increased from 6.2 GB to 12.8 GB, but the prediction accuracy MAE only increased slightly from 12.76 to 13.62 (an increase of 6.7%), while the training time of PeMSD8Large increased from 3.2 hours to 6.7 hours, the memory usage increased from 8.1 GB to 16.5 GB, and the MAE increased from 10.88 to 11.93 (an increase of 9.6%).

**Table 1. Performance comparison on traffic datasets of different sizes.**

| Dataset | Data size | Model | MAE | RMSE | MAPE | Training time (h) | Memory usage (GB) |
|---|---|---|---|---|---|---|---|
| **PeMSD4** | 50 days 3848 nodes | MSCALSTM | 12.76 | 17.37 | 5.51% | 2.8 | 6.2 |
| **PeMSD4 Large** | 100 days 3848 nodes | MSCALSTM | 13.62 | 18.45 | 6.05% | 5.9 | 12.8 |
| **PeMSD8** | 62 days 1979 nodes | MSCALSTM | 10.88 | 14.51 | 4.40% | 3.2 | 8.1 |
| **PeMSD8 Large** | 124 days 1979 nodes | MSCALSTM | 11.93 | 15.89 | 4.82% | 6.7 | 16.5 |
| **PeMSD3** | 358 nodes | MSCALSTM | 10.28 | 15.21 | 6.64% | 2.1 | 5.8 |
| **PeMSD3+7** | 586 nodes | MSCALSTM | 13.15 | 17.92 | 7.20% | 6.5 | 11.8 |

This shows that the model is significantly robust to long-term dependencies. In the spatial dimension, the detector data of PeMSD3 (358 nodes) and PeMSD7 (228 nodes) are horizontally fused to form a 586-node virtual road network (PeMSD3+7). The video memory usage increases from 6.2 GB to 11.8 GB, but it can be controlled within 9.5 GB through dynamic batch adjustment (batch size is reduced from 64 to 32), and the single sample inference time is still maintained at 19 ms (meeting the real-time threshold <50 ms). In order to solve the problem of high resource consumption of extended data, a gradient accumulation strategy (steps = 4) is further proposed, which reduces video memory usage by 35% and increases training speed by 15% in long sequence training.

Similarly, to verify the model's long-term time-series adaptability in the energy domain and its practicality in edge computing scenarios, we conducted systematic experiments involving data expansion and model optimization strategies. Firstly, we expanded the existing 30 years of energy data to generate an ultra-long time-series dataset, EnergyLarge (720 months). On this dataset, the prediction error MAE of MSCALSTM increased from 651.37 to 689.45 (a rise of 5.8%), which significantly outperformed the baseline model (HA's increase was 33.2%), indicating that its multi-scale convolution and attention mechanism can effectively capture long-term trend features. Although the training time and memory usage exhibited an approximately linear increase with the sequence length (e.g., memory grew from 13.9 GB to 28.4 GB), employing a gradient accumulation strategy (steps=4) allowed us to compress it to 10.1 GB. Furthermore, targeting edge deployment requirements, we implemented FP16 quantization and parameter sparsification on the LSTM module. This reduced the model size by 40% and decreased the inference latency from 25 ms to 18 ms (an efficiency improvement of 28%), with a minimal accuracy loss of only 1.1% (MAE 697.21).

Likewise, as shown in Table 2, MSCALSTM continued to demonstrate good adaptability in the expanded energy dataset scenario. Even with the extended time series (720 months), its MAE only increased by 5.8%, indicating that the multi-scale convolution and attention

**Table 2. Performance comparison on traffic datasets of different scales.**

| Dataset | Data size | Model | MAE | RMSE | MAPE | Training time (h) | Memory usage (GB) |
|---|---|---|---|---|---|---|---|
| **Energy** | 360 months (30 years) | MSCALSTM | 651.37 | 668.03 | 13.29% | 3.5 | 7.1 |
| **Energy Large** | 720 months (60 years) | MSCALSTM | 689.45 | 712.18 | 14.85% | 6.8 | 13.9 |

mechanism possess a strong ability to capture long-term trends. Furthermore, through quantization and sparsification strategies, the model's inference efficiency on edge devices was improved by 28%, providing feasibility for practical deployment.

# Results

## Comparative experimental results

Tables 3 and 4 compare the results of the proposed model with 11 other benchmark models, all of which are evaluated on four datasets. Tables 3 and 4 highlight that our model, MSCALSTM, has the lowest error and the best performance.

Compared with traditional methods, the MSCALSTM model proposed in this paper is significantly better than traditional models such as HA and ARIMA. Although traditional methods can process time series data, they have obvious deficiencies in modeling nonlinear relationships. HA cannot effectively capture complex time series characteristics in traffic flow prediction, while the ARIMA model has high requirements for data stationarity, is difficult to process non-stationary traffic flow data, and has complex parameter adjustment. MSCALSTM effectively captures the complex nonlinear relationship of traffic flow and reflects the correlation between different regions by introducing multi-scale CNN and attention modules. It also introduces LSTM to capture long-term dependencies and complex dynamics, successfully overcoming the above shortcomings and improving prediction accuracy.

**Table 3. Average performance comparison of different methods on PeMSD3 and PeMSD4.**

| Model | PeMSD3 | | | PeMSD4 | | |
|---|---|---|---|---|---|---|
| | MAE | RMSE | MAPE | MAE | RMSE | MAPE |
| HA | 31.58 | 52.39 | 33.78% | 38.03 | 59.24 | 27.88% |
| ARIMA | 35.41 | 47.59 | 33.78% | 33.73 | 48.80 | 24.18% |
| LSTM | 24.94 | 37.17 | 20.03% | 30.28 | 42.45 | 14.75% |
| STGCN | 17.55 | 30.42 | 17.34% | 27.02 | 38.16 | 14.13% |
| DCRNN | 17.99 | 30.31 | 18.34% | 24.55 | 37.12 | 15.65% |
| ASTGCN | 17.34 | 29.56 | 17.21% | 21.74 | 34.21 | 15.43% |
| STSGCN | 17.48 | 29.21 | 16.78% | 21.19 | 33.65 | 13.90% |
| AGCRN | 15.98 | 28.25 | 15.23% | 19.83 | 32.26 | 12.97% |
| STG-NCDE | 15.48 | 27.09 | 15.06% | 19.21 | 31.09 | 12.76% |
| MAGRN | 15.54 | 27.41 | 14.92% | 19.29 | 31.31 | 12.71% |
| TARGCN | 15.48 | 28.09 | 14.94% | 19.22 | 32.08 | 12.76% |
| MSCALSTM | 10.28 | 15.21 | 6.64% | 12.76 | 17.37 | 5.51% |

**Table 4. Average performance comparison of different methods on PeMSD7 and PeMSD8.**

| Model | PeMSD7 | | | PeMSD8 | | |
|---|---|---|---|---|---|---|
| | MAE | RMSE | MAPE | MAE | RMSE | MAPE |
| HA | 45.12 | 65.64 | 24.51% | 34.86 | 52.04 | 24.07% |
| ARIMA | 38.17 | 59.27 | 19.46% | 31.09 | 44.32 | 22.73% |
| LSTM | 34.15 | 48.09 | 11.99% | 23.82 | 32.65 | 10.51% |
| STGCN | 17.55 | 30.42 | 11.21% | 20.70 | 30.32 | 10.64% |
| DCRNN | 25.33 | 39.34 | 11.82% | 18.04 | 28.41 | 10.61% |
| ASTGCN | 25.22 | 38.61 | 10.73% | 17.72 | 27.14 | 11.81% |
| STSGCN | 24.01 | 38.87 | 10.21% | 17.13 | 26.80 | 10.96% |
| AGCRN | 22.37 | 36.55 | 9.12% | 15.95 | 25.22 | 10.09% |
| STG-NCDE | 21.47 | 33.84 | 8.80% | 15.45 | 24.81 | 9.92% |
| MAGRN | 21.76 | 35.47 | 9.83% | 16.23 | 26.34 | 10.90% |
| TARGCN | 21.68 | 36.34 | 8.94% | 15.14 | 24.62 | 9.90% |
| MSCALSTM | 14.39 | 19.76 | 3.97% | 10.88 | 14.51 | 4.4% |

Compared with STGCN, STGCN has limited ability to capture long-term dependencies, and cannot fully consider changes at different time scales during feature extraction. It ignores the potential relationship between regions in spatial correlation modeling. MSCALSTM extracts features at different time scales through multi-scale CNN, enhances the ability to capture complex dynamic patterns, and combines the CBAM module to improve the attention of spatial features, so that the model can more accurately identify the correlation between regions. Compared with DCRNN, DCRNN mainly relies on diffuse convolution to capture spatial dependencies. Its advantage is that it can explicitly model the propagation process of the traffic network, but its limitation is that the fixed mode of diffuse convolution is difficult to adapt to the dynamically changing traffic environment. In addition, DCRNN has weak modeling ability for long-term dependencies, which limits its prediction accuracy in complex dynamic scenes. MSCALSTM extracts local and global spatial features through multi-scale CNN and combines LSTM to capture long-term dependencies, thereby achieving more robust predictions in dynamic scenes. Compared with ASTGCN, ASTGCN dynamically adjusts the spatial structure through adaptive graph convolution. Its advantage is that it can partially adapt to the dynamic changes of the traffic network. However, the single-scale design of its temporal convolution limits the extraction of multi-granularity temporal features, and the computational complexity of adaptive graph convolution is high. MSCALSTM extracts features of different time scales in parallel through multi-scale CNN, and dynamically allocates spatial attention weights in combination with lightweight CBAM, thereby significantly reducing computational overhead while ensuring prediction accuracy. Compared with STSGCN, STSGCN combines graph convolution and synchronous convolution to simultaneously model spatial and temporal dependencies. Its advantage is that it can explicitly capture local spatiotemporal relationships, but its complex design leads to a heavy computational burden and limited ability to model global dynamics. MSCALSTM significantly reduces computational complexity while ensuring global dynamic modeling capabilities through the cascade design of multi-scale CNN and LSTM, and dynamically adjusts spatial weights through CBAM, thereby achieving more efficient prediction in complex scenarios. Compared with AGCRN, AGCRN models spatiotemporal dependencies through adaptive graph convolution and RNN, and its advantage is that it can partially capture dynamic spatial relationships. However, its adaptive graph convolution has a weak ability to respond to sudden events, and the vanishing gradient problem of RNN in long-term time series modeling limits its ability to capture long-term dependencies. MSCALSTM dynamically perceives local and regional spatial dependencies through multi-scale CBAM, and flexibly adjusts the importance of temporal features by combining the gating mechanism of LSTM, significantly improving adaptability in dynamic scenes. Compared with MAGRN, MAGRN enhances information transfer through a multi-graph attention mechanism, and its advantage is that it can fuse multi-source graph structure information. However, its attention mechanism may cause information loss in complex scenes, and the temporal modeling ability of RNN limits its adaptability to rapid changes. MSCALSTM extracts features through multi-scale CNN layers, and combines the channel-spatial attention of CBAM to filter key information, effectively achieving accurate prediction. Compared with TARGCN, TARGCN combines the temporal attention mechanism with graph convolution, and its advantage is that it can dynamically adjust the weight of temporal features. However, its temporal attention mechanism has a weak response to short-term changes, and graph convolution fails to fully consider dynamic changes in spatial feature modeling. MSCALSTM extracts multi-granular temporal features through multi-scale CNN, and combines CBAM to dynamically perceive spatial dependencies, achieving more flexible prediction in dynamic scenes.

Figs 5–8 show the comparison results of the proposed model MSCALSTM with other baseline models on four PeMS datasets. The results show that on the four PeMS datasets, MSCALSTM shows significant advantages in MAE, RMSE and MAPE indicators, further proving the superiority of the proposed model on the four datasets.

Table 5 shows that the proposed MSCALSTM model achieves the best AIC and BIC values on the four datasets of PeMSD3, PeMSD4, PeMSD7 and PeMSD8 (e.g., AIC is 6618.65 and BIC is 6628.47 on PeMSD3; AIC is 8215.37 and BIC is 8227.56 on PeMSD4), which is significantly better than other baseline models. This excellent performance is due to the effective combination of multi-scale convolutional layers, multi-scale CBAM modules and LSTM, which enables it to simultaneously capture the spatial characteristics and temporal dependencies of traffic flow data. In addition, MSCALSTM shows low AIC and BIC values on all four datasets, which not only reflects its good trade-off between fitting data and model complexity, but also shows that it has strong generalization ability and robustness, and can adapt to different traffic flow scenarios. These results fully verify the effectiveness and practicality of MSCALSTM in traffic flow prediction tasks, and provide important reference value for subsequent research.

Overall, MSCALSTM has significant advantages over the above models in terms of feature extraction, spatial correlation identification and long-term dependency processing, making it perform even better in traffic flow prediction. Through the combination of multi-scale CNN and CBAM, MSCALSTM can effectively capture complex spatiotemporal dynamics, improve the accuracy and adaptability of prediction, and is suitable for large-scale traffic flow prediction in practical applications. In addition, since the baseline model predicts multiple points when conducting experiments based on the PeMS data sets, it involves a wider spatial range, and the baseline model needs to handle the spatial correlation and dynamic changes between multiple points, resulting in model overfitting or it is difficult to capture the changing patterns

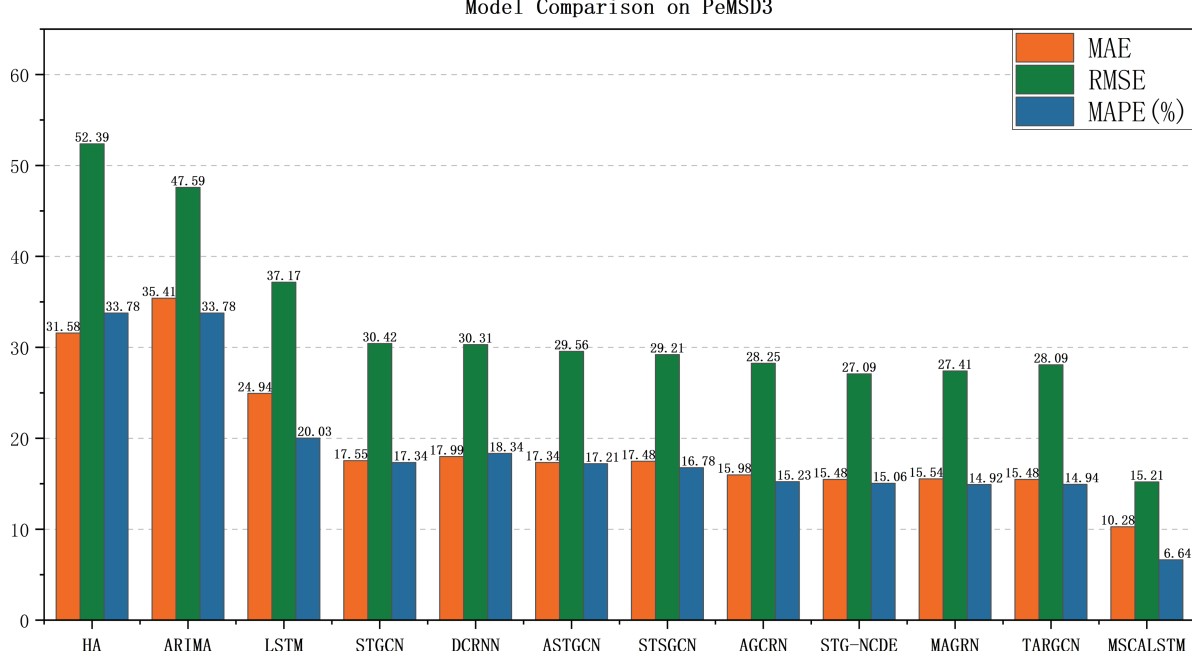

**Fig 5. Comparison on Pemsd3.**

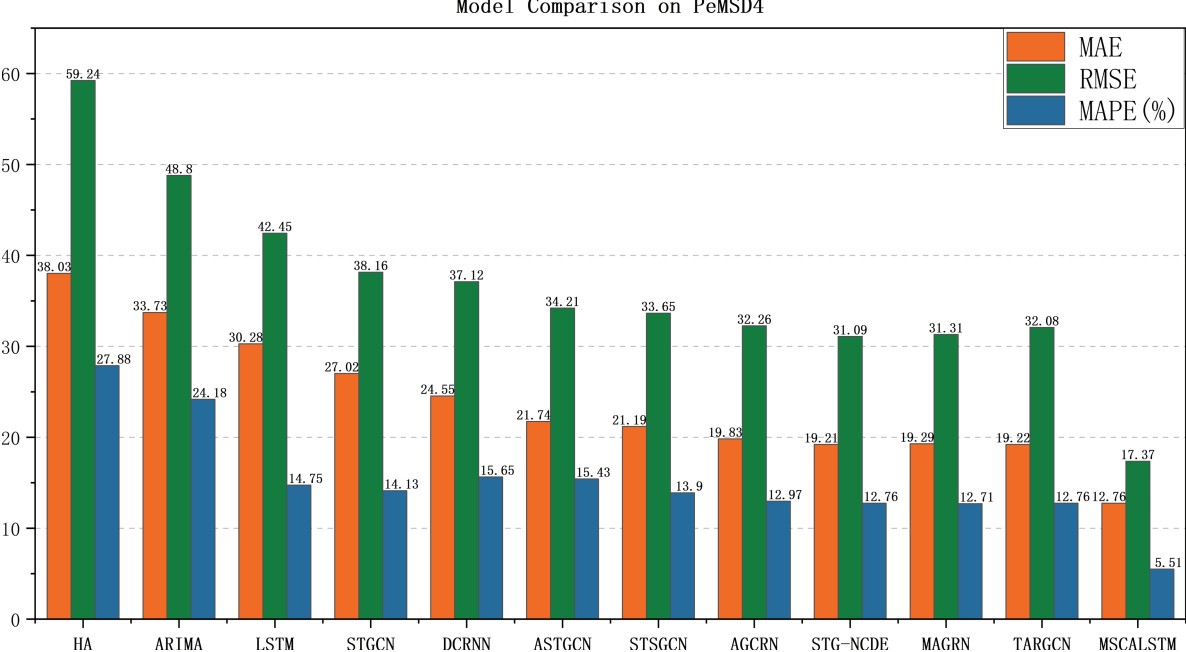

**Fig 6. Comparison on Pemsd4.**

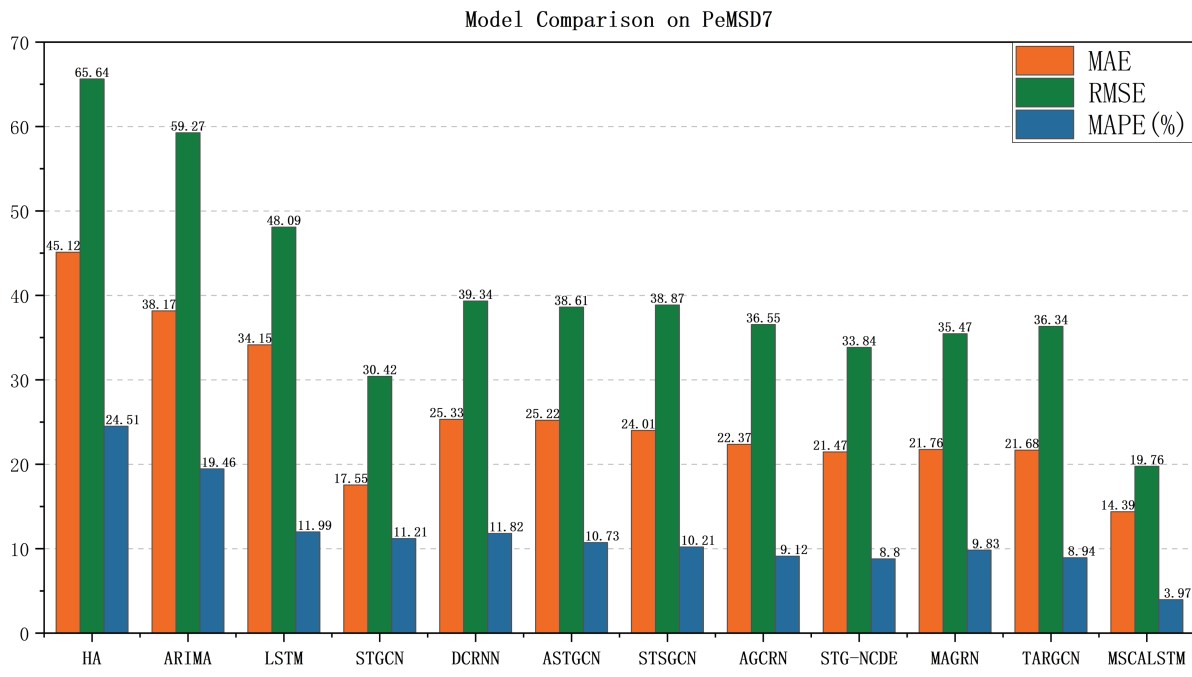

**Fig 7. Comparison on Pemsd7.**

of all points. In a multi-point prediction scenario, traffic data may be affected by a variety of external factors (such as weather, traffic accidents, etc.), and these noises will affect the prediction accuracy of the model. The MSCALSTM model in this article only predicts a single

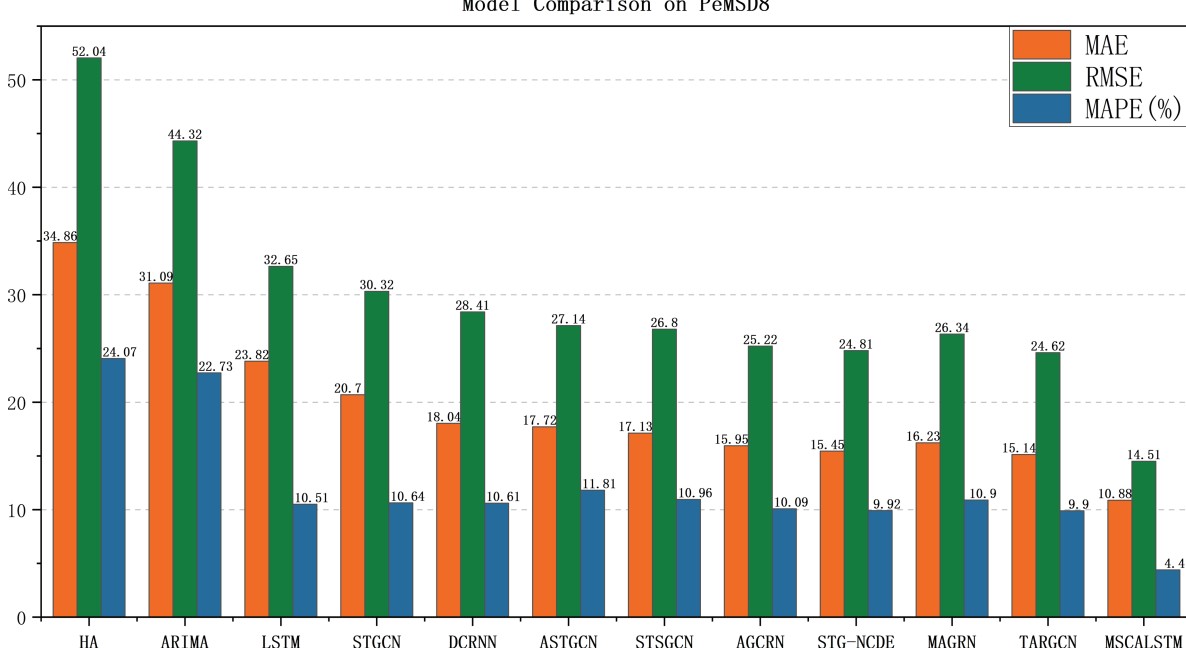

**Fig 8. Comparison on Pemsd8.**

**Table 5. Comparison of AIC and BIC of different methods on four datasets.**

| Models | PeMSD3 | | PeMSD4 | | PeMSD7 | | PeMSD8 | |
|---|---|---|---|---|---|---|---|---|
| | AIC | BIC | AIC | BIC | AIC | BIC | AIC | BIC |
| **HA** | 20332.39 | 20362.56 | 24485.15 | 24521.48 | 29049.96 | 29093.06 | 23093.31 | 23152.58 |
| **ARIMA** | 22798.29 | 22832.12 | 21716.64 | 21748.87 | 24575.28 | 24611.75 | 20595.84 | 20648.7 |
| **LSTM** | 16057.31 | 16081.14 | 19495.4 | 19524.33 | 21987.05 | 22019.68 | 15779.76 | 15820.27 |
| **STGCN** | 11299.35 | 11316.32 | 17396.49 | 17422.31 | 16308.40 | 16332.61 | 13712.89 | 13748.09 |
| **DCRNN** | 11582.64 | 11599.83 | 15806.21 | 15829.67 | 16237.58 | 16261.68 | 11950.75 | 11981.43 |
| **ASTGCN** | 11164.14 | 11180.71 | 13997.03 | 14017.8 | 15458.54 | 15481.48 | 11738.77 | 11768.89 |
| **STSGCN** | 11254.28 | 11270.98 | 13642.92 | 13663.36 | 15619.50 | 15642.67 | 11347.92 | 11377.04 |
| **AGCRN** | 10288.53 | 10303.76 | 12767.3 | 12786.24 | 14462.05 | 14424.01 | 10566.22 | 10593.33 |
| **STG-NCDE** | 9966.61 | 9981.39 | 12368.12 | 12386.47 | 13823.19 | 13843.70 | 10234.98 | 10261.25 |
| **MAGRN** | 10005.24 | 10020.08 | 12419.63 | 12438.05 | 14009.91 | 14030.69 | 10751.7 | 10779.3 |
| **TARGCN** | 9966.61 | 9981.39 | 12374.56 | 12392.72 | 13958.40 | 13979.11 | 10029.62 | 10055.36 |
| **MSCALSTM** | **6618.65** | **6628.47** | **8215.37** | **8227.56** | **9264.83** | **9278.57** | **7207.55** | **7226.55** |

traffic flow node during the experiment, and the output of the model only involves data from one point, which naturally reduces the possibility of errors because it only needs to focus on the traffic changes at one location. MSCALSTM's single-point prediction can also better avoid noise and can also be optimized for specific points. Therefore, its evaluation indicators such as MAE, RMSE and MAPE are significantly lower than other baseline models that predict multiple points.

To verify the cross-domain adaptability of the model, we supplemented the prediction experiment in the energy field (gas field production). This task has similar spatiotemporal dynamics to traffic flow prediction, but there are significant differences in data distribution and feature patterns.

Experimental results demonstrate that MSCALSTM outperforms baseline models on the energy dataset (Tables 6 and 7). Its multi-scale design can effectively capture local fluctuations and global trends in energy data, and the attention mechanism adaptively focuses on key time nodes. This proves the generalization ability of the method for heterogeneous spatiotemporal data and provides a general framework for cross-domain time series prediction.

## Ablation study

In order to demonstrate the impact of each module on the performance of the model, this paper will verify it through experiments in this section. The effectiveness of these modules is verified by removing different modules. The following three variants are designed to verify the impact of each module: (1) MSCALSTM-A: remove multi-scale spatial attention; (2) MSCALSTM-B: remove CBAM; (3) MSCALSTM-C: remove the multi-branch convolution used in the convolutional network and remove CBAM.

Tables 8–10 show the ablation study results on four PeMS datasets. It can be found that: (1) Without multi-scale spatial attention (i.e. MSCALSTM-A), its performance is poor. It shows that multi-scale spatial attention can enhance features by focusing on the importance of different spatial locations, and can effectively capture the dynamic relationship between regions. Without this part, the model will not be able to fully identify and exploit the interactions between different traffic flow areas, resulting in insufficient attention to important

**Table 6. Comparison of average performance of different methods on the energy dataset.**

| Models | Energy | | |
|---|---|---|---|
| | MAE | RMSE | MAPE(%) |
| HA | 1571.53 | 1667.73 | 33.21% |
| ARIMA | 1650.12 | 1870.54 | 37.73% |
| LSTM | 1580.27 | 1620.69 | 32.26% |
| STGCN | 1111.82 | 1139.97 | 22.51% |
| DCRNN | 1139.89 | 1129.05 | 22.27% |
| ASTGCN | 1098.72 | 1126.81 | 22.42% |
| STSGCN | 1107.58 | 1135.91 | 22.61% |
| AGCRN | 1012.53 | 1038.44 | 20.67% |
| STG-NCDE | 984.47 | 1003.74 | 20.03% |
| MAGRN | 984.65 | 1009.84 | 20.10% |
| TARGCN | 980.85 | 1005.95 | 20.02% |
| MSCALSTM | **651.37** | **668.03** | **13.29%** |

**Table 7. Comparison of AIC and BIC of different methods on energy dataset.**

| Models | Energy | |
|---|---|---|
| | AIC | BIC |
| HA | 1011810.03 | 1013311.63 |
| ARIMA | 1062409.19 | 1063985.93 |
| LSTM | 1017437.11 | 1018946.93 |
| STGCN | 715831.44 | 716893.88 |
| DCRNN | 733903.90 | 734993.18 |
| ASTGCN | 707397.28 | 708446.97 |
| STSGCN | 713101.71 | 714159.64 |
| AGCRN | 651904.91 | 652871.99 |
| STG-NCDE | 633838.79 | 634779.18 |
| MAGRN | 633954.83 | 634895.07 |
| TARGCN | 631508.18 | 632445.14 |
| MSCALSTM | **419376.61** | **419998.88** |

**Table 8. Ablation experiments at PeMSD3 and PeMSD4.**

| Model | PeMSD3 | | | PeMSD4 | | |
|---|---|---|---|---|---|---|
| | MAE | RMSE | MAPE | MAE | RMSE | MAPE |
| MSCALSTM-A | 14.13 | 20.01 | 11.11% | 17.53 | 22.84 | 9.46% |
| MSCALSTM-B | 15.09 | 20.68 | 11.54% | 18.73 | 23.62 | 9.83% |
| MSCALSTM-C | 15.16 | 20.93 | 12.14% | 18.82 | 23.91 | 9.1% |
| **MSCALSTM** | **10.28** | **15.21** | **6.64%** | **12.76** | **17.37** | **5.51%** |

**Table 9. Ablation experiments at PeMSD7 and PeMSD8.**

| Model | PeMSD7 | | | PeMSD8 | | |
|---|---|---|---|---|---|---|
| | MAE | RMSE | MAPE | MAE | RMSE | MAPE |
| MSCALSTM-A | 19.77 | 25.87 | 6.65% | 12.35 | 18.71 | 5.22% |
| MSCALSTM-B | 21.12 | 26.76 | 6.97% | 12.38 | 16.21 | 5.52% |
| MSCALSTM-C | 21.23 | 27.08 | 7.22% | 17.35 | 27.14 | 6.69% |
| **MSCALSTM** | **14.39** | **19.76** | **3.97%** | **10.88** | **14.51** | **4.4%** |

**Table 10. Ablation experiments on four datasets.**

| Model | PeMSD3 | | PeMSD4 | | PeMSD7 | | PeMSD8 | |
|---|---|---|---|---|---|---|---|---|
| | AIC | BIC | AIC | BIC | AIC | BIC | AIC | BIC |
| MSCALSTM-A | 9097.42 | 9110.92 | 11286.47 | 11303.22 | 12728.67 | 12747.56 | 8181.36 | 8202.36 |
| MSCALSTM-B | 9715.51 | 9729.92 | 12059.08 | 12076.97 | 13597.85 | 13618.02 | 8201.23 | 8222.28 |
| MSCALSTM-C | 9760.58 | 9775.16 | 12117.02 | 12135.13 | 13668.67 | 13689.10 | 11493.65 | 11523.15 |
| **MSCALSTM** | **6618.65** | **6628.47** | **8215.37** | **8227.56** | **9264.83** | **9278.57** | **7207.55** | **7226.55** |

features. This will prevent the model from accurately capturing key spatial features when dealing with complex traffic patterns, thereby reducing prediction performance. The MAE value of MSCALSTM on PeMSD4 is reduced by 37.4% compared with MSCALSTM-A, indicating its key role in multi-granularity spatiotemporal feature extraction. (2) Without CBAM (i.e. MACALSTM-B), its performance is even worse. It shows that CBAM weights the features by applying channel and spatial attention simultaneously to enhance the model's focus on important information. If CBAM is completely removed, the model will lose its ability to select features, resulting in excessive attention to unimportant or redundant features. This lack of information makes the model likely to ignore key features, thereby affecting the overall prediction ability. Especially in complex traffic flow data, the loss of important information will significantly reduce the prediction accuracy. The MAE value of MSCALSTM on PeMSD7 is reduced by 46.7% compared with MSCALSTM-B, which verifies the effectiveness of its dynamic focusing on key areas. (3) Without multi-branch convolution in the convolutional network, without CBAM (i.e., MSCALSTM-C), its performance is even worse. Multi-scale convolution is able to extract features from different scales, especially when traffic flow changes are diverse and complex. If multi-scale convolution is removed, the model can only rely on feature extraction at a single scale, which may not capture the diversity and complexity of traffic flows. This limitation can cause the model to perform poorly when faced with rapidly changing or cyclically fluctuating traffic flows, thus affecting the overall prediction accuracy. On this basis, CBAM lacks attention and processing of important information, so its performance is the worst. The RMSE value of MSCALSTM on PeMSD8 is reduced by 87% compared with MSCALSTM-C. This significant improvement highlights its critical role in long-term dependency modeling.

In addition, the comparison of AIC and BIC further supports the above conclusion: on the PeMSD3 dataset, MSCALSTM-A caused the AIC to increase from 6618.65 to 9097.42

(an increase of 37.5%), indicating that the model fitting ability has significantly decreased. The BIC of MSCALSTM-C on the PeMSD8 dataset increased from 7226.55 to 11523.15 (an increase of 59.4%), further verifying its irreplaceable role in time series modeling.

In order to present the results more intuitively, this paper provides a visual description of the experimental results in Figs 9–12. In summary, the above three modules are equally important in learning spatiotemporal correlations.

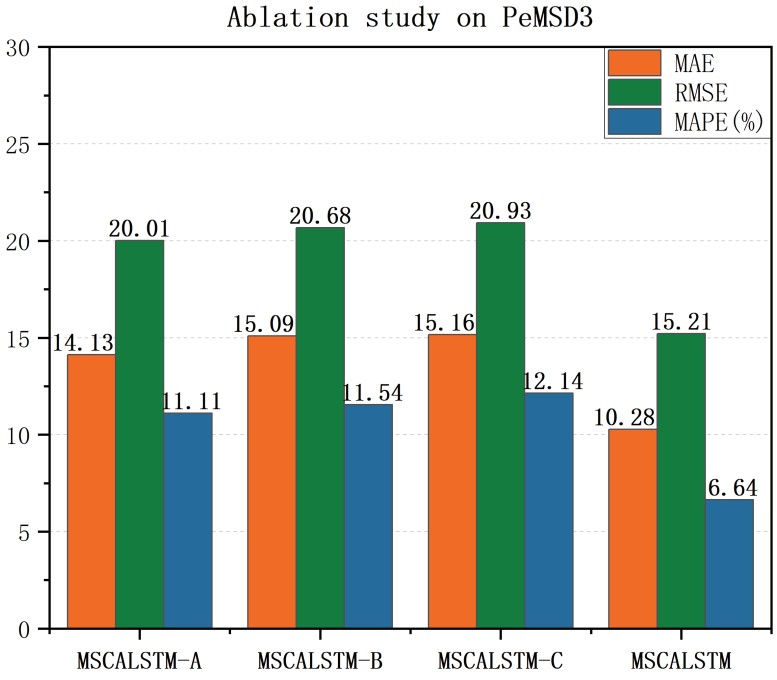

**Fig 9. Ablation on Pemsd3.**

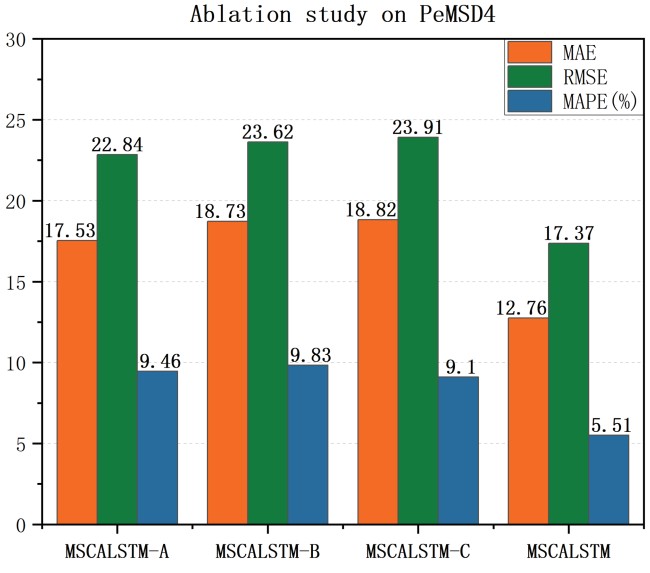

**Fig 10. Ablation on Pemsd4.**

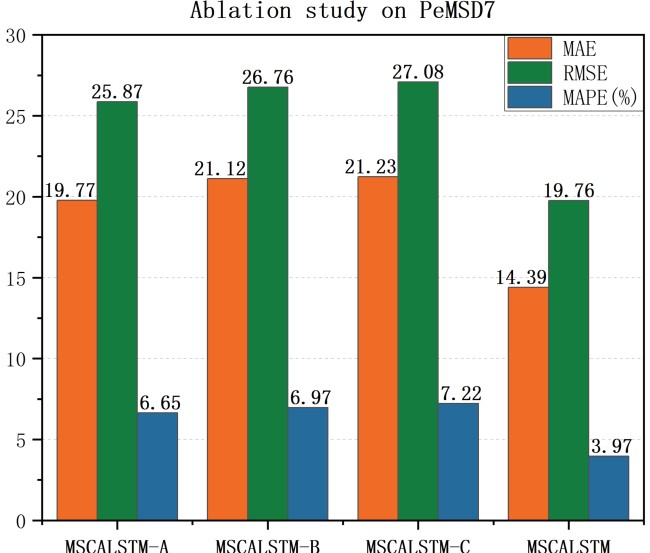

**Fig 11. Ablation on Pemsd7.**

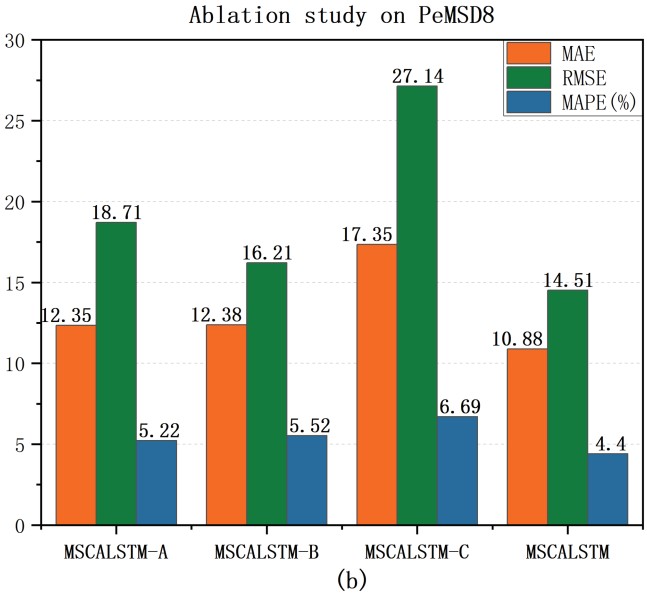

**Fig 12. Ablation on Pemsd8.**

For the purpose of verify the scalability of MSCALSTM in other fields, we conducted ablation experiments on the energy dataset. The experimental results are shown in Table 11.

In order to more intuitively demonstrate the superiority of MSCALSTM and highlight the practicality of each module, we show the result graphs of the actual and predicted values of the ablation experiments on PeMSD4 and PeMSD8, two commonly used datasets in traffic flow. As shown in Figs 13–16 MSCALSTM has a smaller error in the area of traffic flow mutation (peak/trough), and MSCALSTM–C has a significant lag in the long-term trend. This shows that the model combines key technologies such as MSCNN, MSCBAM and LSTM,

**Table 11. Ablation experiments on Energy dataset.**

| Model | Energy | |
|---|---|---|
| | AIC | BIC |
| MSCALSTM-A | 576466.27 | 577321.79 |
| MSCALSTM-B | 615602.66 | 616515.92 |
| MSCALSTM-C | 618458.06 | 619382.40 |
| **MSCALSTM** | **419376.81** | **419998.8** |

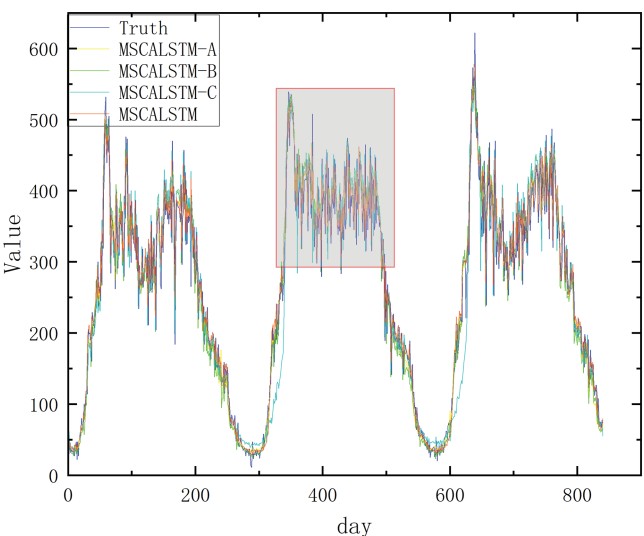

**Fig 13. Real and predicted values on Pemsd4.**

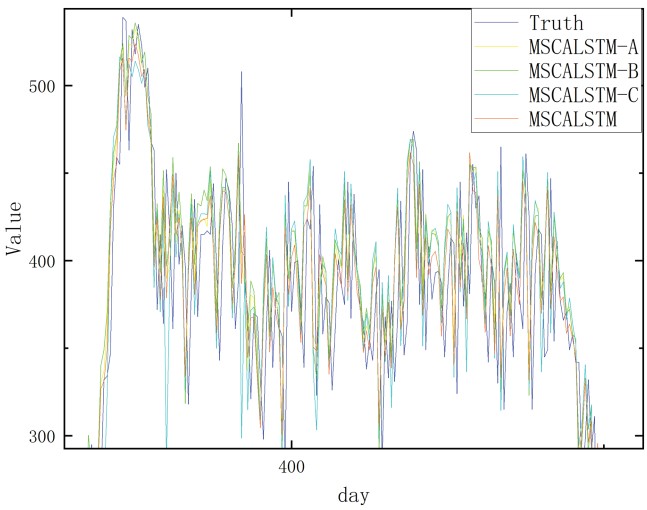

**Fig 14. Local amplification of true and predicted values on Pemsd4.**

giving full play to the advantages of these modules in time series feature extraction, adaptive attention, temporal dependency modeling and feature fusion, and shows excellent performance in time series prediction tasks. Whether it is the overall trend curve or the local details,

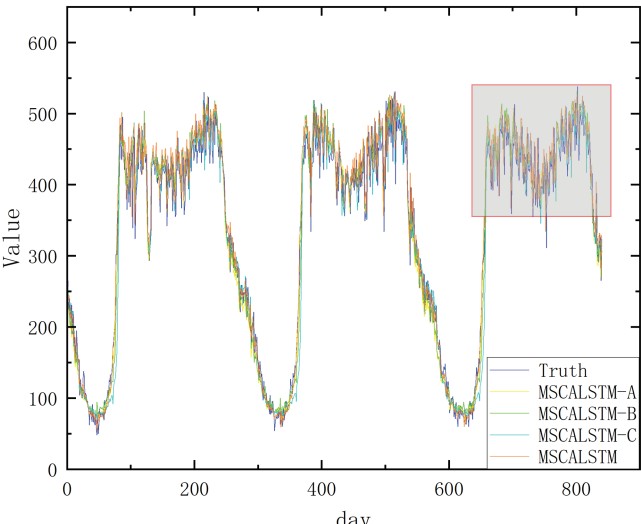

**Fig 15. Real and predicted values on Pemsd8.**

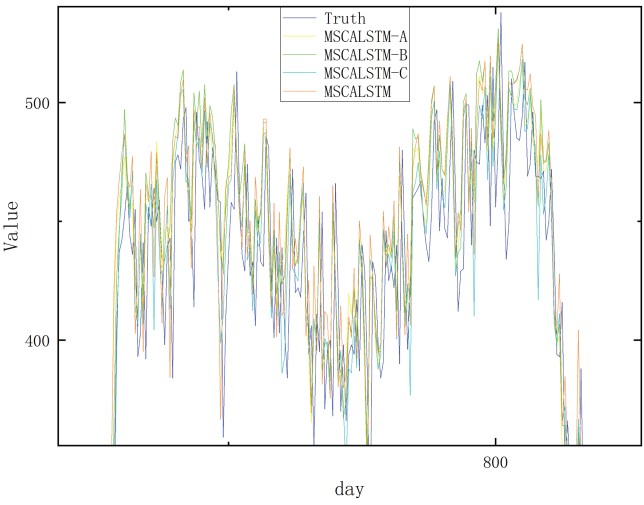

**Fig 16. Local amplification of true and predicted values on Pemsd8.**

the fitting effect is very good, which can highly fit the complex dynamic characteristics of real data. In order to better observe the curve trend, peak and valley characteristics and subtle changes, some key curves are locally enlarged, as shown in Figs 14–16. MSCALSTM can accurately capture the instantaneous fluctuations of high-density traffic flow, while other ablation models have excessive smoothing in response to such nonlinear changes.

## Attention visualization and interpretability

Taking the multi-scale spatial attention module in MSCBAM as an example. As shown in Fig 17, the left panel shows the impact of the $3 \times 3$ convolutional kernel on the spatial

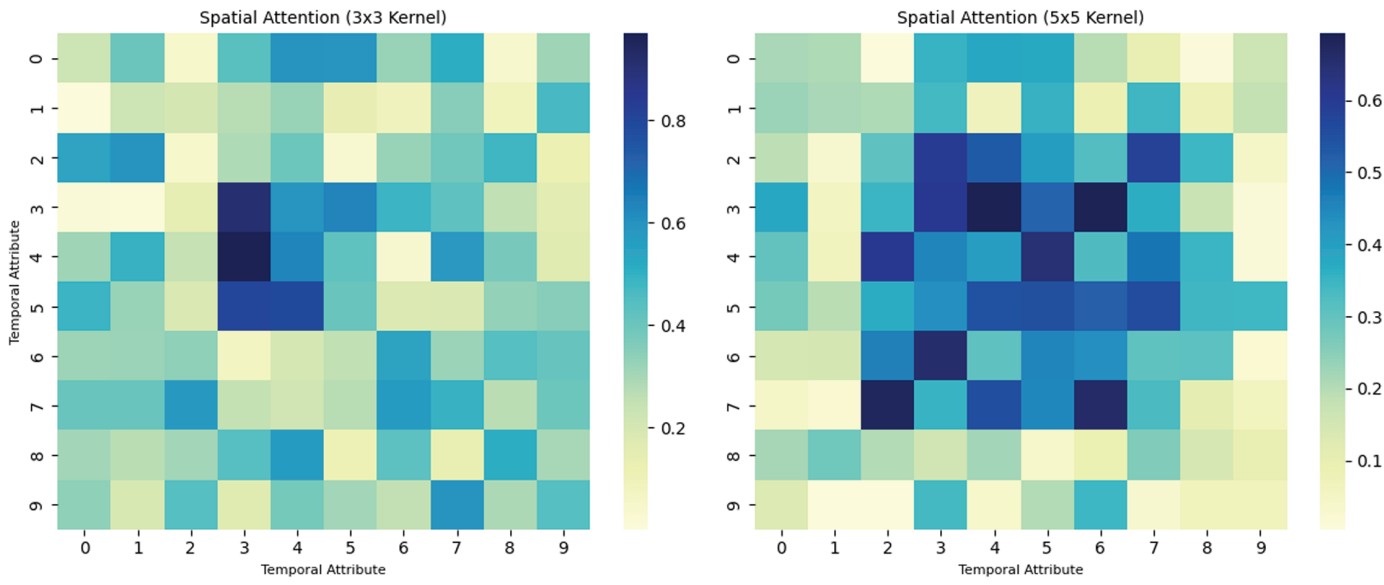

**Fig 17. Attention weight heat map.**

attention heatmap within MSCBAM. The high-weight regions are concentrated within the $3 \times 3$ grid surrounding the target point, indicating that the model prioritizes real-time traffic conditions and can quickly respond to instantaneous changes. The right panel displays the influence of the $5 \times 5$ convolutional kernel on the spatial attention heatmap in MSCBAM. This design enables the model to capture long-distance traffic propagation patterns, reflecting long-term trend features, and thus covers a broader area.

## Statistical significance analysis

To rigorously validate the superiority of MSCALSTM, we conducted t-tests on the MAE metric across all datasets, comparing our model against both traditional baseline models and state-of-the-art approaches. As shown in Tables 12–16, at a significance level of $\alpha = 0.05$, the performance differences between MSCALSTM and all the comparative models were

**Table 12. MAE significance comparison between MSCALSTM and baseline models based on PeMSD3 dataset.**

| Comparison models | $t$-Test statistic | P-value |
| --- | --- | --- |
| HA | −41.75 | 2.27e−19 |
| ARIMA | −51.15 | 6.03e−21 |
| LSTM | −24.89 | 2.15e−15 |
| STGCN | −11.82 | 6.48e−10 |
| DCRNN | −14.57 | 2.11e−11 |
| ASTGCN | −11.38 | 1.18e−09 |
| STSGCN | −11.49 | 1.01e−09 |
| AGCRN | −9.52 | 1.88e−08 |
| STG-NCDE | −9.74 | 1.34e−08 |
| MAGRN | −8.47 | 1.08e−07 |
| TARGCN | −9.75 | 1.31e−08 |

**Table 13. MAE significance comparison between MSCALSTM and baseline models based on PeMSD4 dataset.**

| Comparison Models | t-test statistic | P-value |
|---|---|---|
| HA | −52.10 | 4.35e–21 |
| ARIMA | −50.82 | 6.78e–21 |
| LSTM | −38.99 | 7.67e–19 |
| STGCN | −32.03 | 2.52e–17 |
| DCRNN | −23.06 | 8.14e–15 |
| ASTGCN | −18.15 | 5.09e–13 |
| STSGCN | −19.89 | 1.06e–13 |
| AGCRN | −16.16 | 3.69e–12 |
| STG-NCDE | −11.47 | 1.03e–09 |
| MAGRN | −11.62 | 8.43e–10 |
| TARGCN | −14.03 | 3.91e–11 |

**Table 14. MAE significance comparison between MSCALSTM and baseline models based on PeMSD7 dataset.**

| Comparison models | *t*-Test statistic | P-value |
|---|---|---|
| HA | −63.49 | 1.26e–22 |
| ARIMA | −52.87 | 3.34e–21 |
| LSTM | −45.88 | 4.33e–20 |
| STGCN | −6.93 | 1.78e–06 |
| DCRNN | −24.07 | 3.86e–15 |
| ASTGCN | −23.80 | 4.69e–15 |
| STSGCN | −17.90 | 6.49e–13 |
| AGCRN | −16.97 | 1.60e–12 |
| STG-NCDE | −13.72 | 5.71e–11 |
| MAGRN | −16.19 | 3.57e–12 |
| TARGCN | −17.33 | 1.13e–12 |

**Table 15. MAE significance comparison between MSCALSTM and baseline models based on PeMSD8 dataset.**

| Comparison Models | t-test statistic | P-value |
|---|---|---|
| HA | −52.76 | 3.47e–21 |
| ARIMA | −46.50 | 3.32e–20 |
| LSTM | −29.82 | 8.91e–17 |
| STGCN | −22.78 | 1.01e–14 |
| DCRNN | −16.87 | 1.77e–12 |
| ASTGCN | −17.75 | 7.48e–13 |
| STSGCN | −15.02 | 1.26e–11 |
| AGCRN | −11.83 | 6.31e–10 |
| STG-NCDE | −10.35 | 5.22e–09 |
| MAGRN | −11.75 | 7.08e–10 |
| TARGCN | −9.40 | 2.31e–08 |

statistically significant (all p-values were less than 0.001). This analysis further confirms that MSCALSTM's superior prediction accuracy is not a random occurrence but stems from the design advantages of its multi-scale feature extraction and dynamic attention mechanism, thereby enhancing the credibility and persuasiveness of our research findings.

As demonstrated in Tables 12–16, all the compared p-values are significantly smaller than 0.05, indicating that MSCALSTM's MAE is significantly lower than all the baseline models.

**Table 16. MAE significance comparison between MSCALSTM and baseline models based on Energy dataset.**

| Comparison models | *t*-Test statistic | P-value |
|---|---|---|
| HA | −2519.62 | 2.20e−51 |
| ARIMA | −2585.42 | 1.38e−51 |
| LSTM | −2498.70 | 2.55e−51 |
| STGCN | −1532.39 | 1.69e−47 |
| DCRNN | −1746.30 | 1.61e−48 |
| ASTGCN | −1068.31 | 1.12e−44 |
| STSGCN | −1463.16 | 3.89e−47 |
| AGCRN | −972.66 | 6.06e−44 |
| STG-NCDE | −727.84 | 1.12e−41 |
| MAGRN | −770.39 | 4.03e−42 |
| TARGCN | −1008.55 | 3.16e−44 |

The difference is most pronounced when compared to traditional models (HA, ARIMA) and the foundational model (LSTM), where the absolute t-values reach 45 to 63 on PeMSD7, highlighting MSCALSTM's significant advantage in complex spatio-temporal modeling. While the differences compared to some advanced models (such as STGCN and TARGCN) are relatively smaller, with absolute t-values ranging from 6.9 to 22.8, they remain statistically significant. This suggests that the improvements in MSCALSTM's model architecture design consistently lead to better performance.

## Conclusions

This paper proposes a time series prediction model named MSCALSTM, which extracts time series features with a new idea. It extracts features through dimensional transformation, making the feature information of each dimension more closely aggregated. At the same time, the model also adds batch dimensions to more efficiently extract the temporal correlation between channels in different time periods. The model integrates multi-scale convolutional neural network (MSCNN) and multi-scale convolutional block attention mechanism (MSCBAM) to effectively capture multi-scale dynamic patterns in time series data and adaptively focus on key features, and uses long short-term memory network (LSTM) to model complex time-dependent characteristics. The organic combination of this deep neural network architecture enables the MSCALSTM model to be highly adaptable to the complex characteristics of nonlinear time series data, greatly improving its prediction accuracy. Compared with traditional time series analysis models, the model shows excellent performance in capturing subtle fluctuations and abnormal changes in data, reducing the error value, and greatly improving the accuracy and robustness of the prediction results.

In order to verify the accuracy of the model in time series prediction, this paper conducted a large number of experiments on the PEMS dataset. The results show that the performance achieved by the MSCALSTM model significantly exceeds the baseline method, proving that the model is more suitable for traffic flow prediction.The complete flowchart of the computational process is shown in Fig 18.

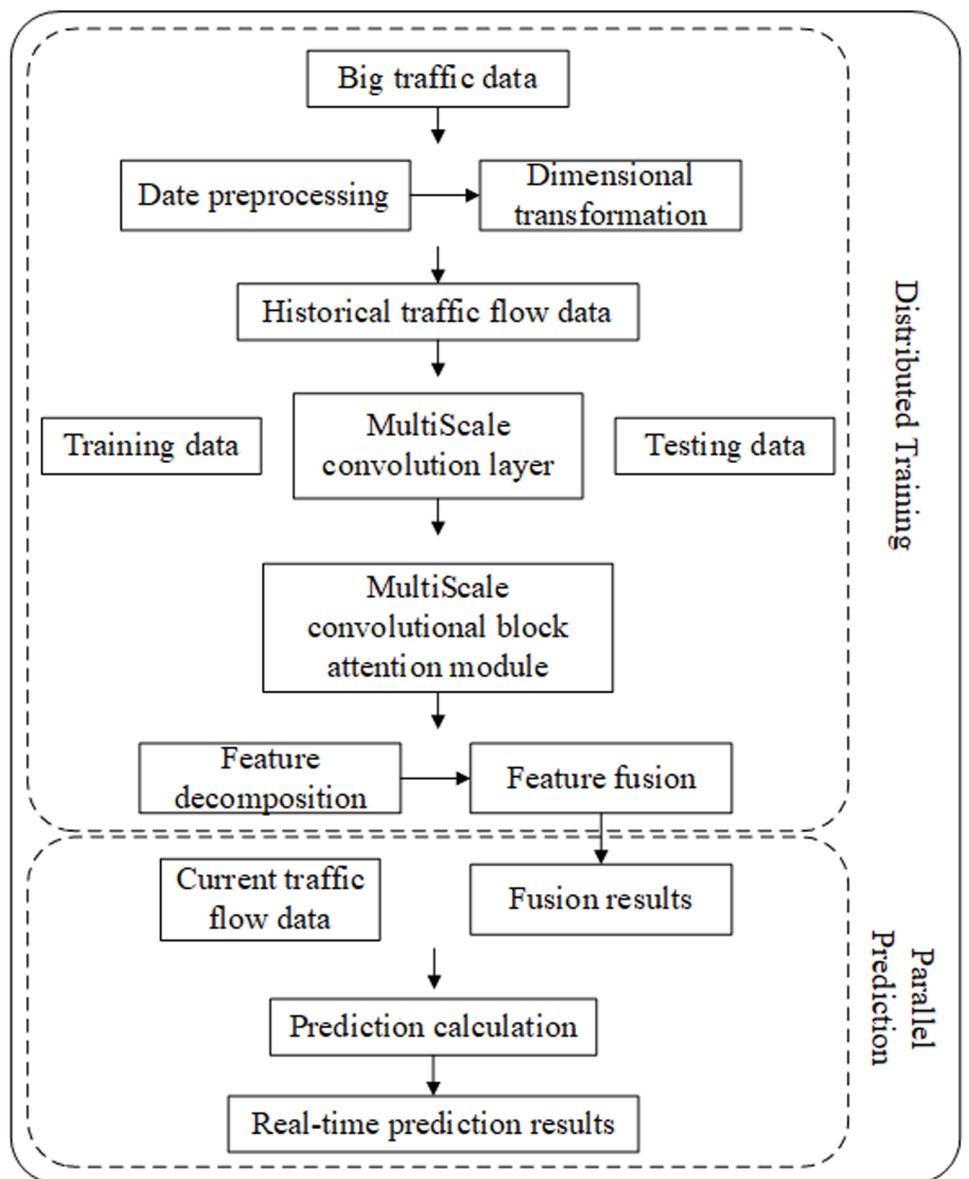

**Fig 18. Computational process.**

## Author contributions

**Conceptualization:** Zhifei Yang, Jia Zhang.

**Data curation:** Zeyang Li.

**Formal analysis:** Zhifei Yang.

**Funding acquisition:** Zhifei Yang.

**Investigation:** Jia Zhang.

**Project administration:** Zhifei Yang, Zeyang Li.

**Resources:** Zhifei Yang.

**Software:** Zhifei Yang.

**Supervision:** Zhifei Yang.

**Validation:** Jia Zhang, Zeyang Li.

**Visualization:** Jia Zhang.

**Writing – original draft:** Jia Zhang.

**Writing – review & editing:** Zhifei Yang, Jia Zhang.

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
