## [Decision Letter · Decision Letter 0]

27 Dec 2024

PONE-D-24-51425Multi-scale time series prediction model based on deep learning and its applicationPLOS ONE

Dear Dr. Yang,

Thank you for submitting your manuscript to PLOS ONE. After careful consideration, we feel that it has merit but does not fully meet PLOS ONE’s publication criteria as it currently stands. Therefore, we invite you to submit a revised version of the manuscript that addresses the points raised during the review process.

Major revision: Please address the comments by the reviewers.

We look forward to receiving your revised manuscript.

Kind regards,

Zeyar Aung

Academic Editor

PLOS ONE

Journal Requirements:

“The Humanities and Social Sciences Research Project of the Ministry of Education (20YJCZH212)

The Higher Education Research Project of the Education Department of Gansu Province (2020B-115)

The Experimental Teaching Reform Project of Lanzhou Jiao tong University (2024013).”

“This research is supported by the Humanities and Social Sciences Research Project of the Ministry of Education (20YJCZH212), the Higher Education Research Project of the Education Department of Gansu Province (2020B-115), and the Experimental Teaching Reform Project of Lanzhou Jiao tong University (2024013)."

“The Humanities and Social Sciences Research Project of the Ministry of Education (20YJCZH212)

The Higher Education Research Project of the Education Department of Gansu Province (2020B-115)

The Experimental Teaching Reform Project of Lanzhou Jiao tong University (2024013).”

Reviewers' comments:

Reviewer's Responses to Questions

**Comments to the Author**

1. Is the manuscript technically sound, and do the data support the conclusions?

Reviewer #1: Partly

Reviewer #2: Yes

2. Has the statistical analysis been performed appropriately and rigorously? 

Reviewer #1: N/A

Reviewer #2: No

3. Have the authors made all data underlying the findings in their manuscript fully available?

Reviewer #1: No

Reviewer #2: Yes

4. Is the manuscript presented in an intelligible fashion and written in standard English?

Reviewer #1: Yes

Reviewer #2: Yes

5. Review Comments to the Author

Reviewer #1: Report

Manuscript Number: PONE-D-24-51425

Article Type: Research Article

Title: Multi-scale time series prediction model based on deep learning and its application

The manuscript proposes a deep learning-based multi-scale time series prediction model (MSCALSTM) aimed at enhancing the accuracy and robustness of traffic flow forecasting. The paper points out that traditional time series prediction models, such as Long Short-Term Memory networks (LSTM) and Convolutional Neural Networks (CNN), have limitations in dealing with complex non-linear time dependencies and capturing the intricate characteristics of traffic flow data. To overcome these limitations, the authors propose an integrated model that combines a Multi-Scale Convolutional Neural Network (MSCNN), a Multi-Scale Convolution Block Attention Module (MSCBAM), and LSTM. The MSCNN is responsible for capturing multi-scale dynamic patterns in time series data, the MSCBAM adaptively focuses on key features, and the LSTM excels at modeling complex temporal dependencies. This integrated approach leverages the strengths of various techniques, effectively improving the accuracy and robustness of time series predictions. The performance of the MSCALSTM model is validated through experiments on two real-world datasets from the California Transportation Performance Measurement System (PEMS). The experimental results show that MSCALSTM significantly outperforms existing state-of-the-art models in terms of prediction accuracy. Although the paper shows some interesting results, there are still some issues that need to be addressed.

1. The authors mentioned four contributions of their research in the introduction; it is recommended that they further enhance the discussion.

2. Although the paper uses datasets from the California Transportation Performance Measurement System (PEMS) for experimental validation, these datasets may not fully represent the diversity of traffic flow globally. The authors could consider testing the model with datasets from more diverse regions and conditions to enhance the model's generalization and adaptability. Alternatively, the authors could analyze the predictive performance of the proposed method under different data lengths to further strengthen robustness analysis.

3. The paper lacks a detailed description of the model's hyperparameter selection and adjustment process. Further analysis and clarification are recommended.

4. Although the model integrates an attention mechanism, its internal working mechanisms and decision processes may not be transparent enough for end-users. It is suggested to strengthen the discussion and analysis.

5. The evaluation metrics in the paper are overly simplistic; it is recommended to adopt more loss functions or statistical tests such as AIC, BIC, etc., to further test the model's performance.

6. The authors' figures are not clear enough, especially the last two images which are difficult to recognize; please enhance them.

7. For their proposed method, it is recommended to provide a computational流程 or pseudocode analysis.

Reviewer #2: The abstract

The abstract introduces several abbreviations, such as MSCNN and MSCALSTM, without adequately defining them or providing sufficient context, which may hinder the reader's understanding.

Introduction and related work

Strengths:

Clear Context and Motivation: The introduction effectively establishes the importance of addressing traffic congestion in urban areas and frames traffic flow prediction as a cost-effective solution.

References: It provides a comprehensive overview of traditional and state-of-the-art methods, including statistical, machine learning, and deep learning techniques, with relevant citations.

Clear Problem Statement: The limitations of existing models, including overfitting, limited scalability, and the challenges of manually designed attention mechanisms, are clearly articulated.

Weaknesses:

Repetition and Redundancy

Several points are repeated unnecessarily, such as the advantages of CNN and LSTM, or the improvement in CBAM using multi-scale convolutional layers.

Example: “Using convolutions of different scales can enable the module to better model the spatial relationship between features” appears twice.

Literature Review

The overall review of the state-of-the-art methods is relevant and comprehensive, covering traditional statistical models, machine learning techniques, and deep learning approaches. However, I was surprised to see that the combination of wavelet transforms with machine learning models was mentioned only once and that Fourier transform-based approaches combined with nonlinear statistical models were not highlighted. For instance, relevant examples include:

Wavelet and machine learning combinations for predictive modeling (https://doi.org/10.1016/j.compeleceng.2024.109631).

Fourier transform applications in air quality forecasting (https://doi.org/10.1016/j.atmosenv.2023.120210).

Spectral band decomposition with nonlinear models for time series prediction, as in Ouaret et al., Stochastic Environmental Research and Risk Assessment (https://doi.org/10.1007/s00477-017-1510-).

Incorporating a discussion on these methods could enrich the literature review by addressing additional perspectives on time series forecasting techniques.

Overwhelming Length

The introduction is dense and too detailed, particularly in the review of related work. While the level of detail may be suitable for the related work section, condensing it in the introduction would improve readability.

Focus and Structure

The introduction tries to cover too many aspects, from statistical methods to recent developments in attention mechanisms, which can dilute the focus.

The narrative could be more streamlined to emphasize the motivation, limitations, and the novelty of the proposed approach.

Clarity in Terminology and Abbreviations

Abbreviations like ARIMA, MSCALSTM, MSCNN, and MSCBAM are introduced with little explanation, which can confuse readers unfamiliar with the terms.

Avoid introducing excessive technical terms in the introduction unless necessary.

Minor Language Issues

Some sentences are awkward or unclear. For example: “Due to the nonlinear characteristics of traffic flow data, overfitting is common in traditional machine learning methods, overfitting is common in traditional machine learning methods...”

Suggestions for Improvement:

Condense the Literature Review

Focus on the most relevant advancements, such as the transition from statistical methods to deep learning approaches. Move details about individual methods to the related work section.

Streamline and Eliminate Redundancy

Avoid repeating points about multi-scale convolutional layers and the advantages of CNN-LSTM combinations.

Simplify Abbreviation Introductions

Briefly explain MSCALSTM, MSCNN, and MSCBAM when first introduced and use them consistently throughout.

Reorganize for Flow

Start with the problem of traffic congestion and the importance of traffic flow prediction, then outline existing challenges, and conclude with the novelty of the proposed solution.

Refine Language and Remove Errors

Fix repetitive phrases and improve sentence clarity. For example, the sentence about overfitting in machine learning methods could be rewritten as:

“Traditional machine learning methods often suffer from overfitting due to the nonlinear characteristics of traffic flow data, which impacts prediction performance.”

Keep the Contributions Section Concise

Avoid repeating the same idea within the contributions, and rephrase for precision.

Method Overview and Experimental Analysis

The methodology of MSCALSTM demonstrates several strengths that make it a compelling contribution. It offers a well-structured breakdown of the model into three main modules—MSCNN, MSCBAM, and LSTM—each tailored to address specific challenges in time series prediction. The multi-scale convolutional approach (MSCNN) is particularly notable for its ability to capture features at varying scales using convolution kernels of sizes 1×1, 3×3, and 5×5, which enhances feature richness, expands the receptive field, and improves prediction accuracy. Additionally, the inclusion of a multi-scale attention mechanism (MSCBAM) adapted from CBAM effectively emphasizes key features and adaptively focuses on important areas, improving both interpretability and performance. The integration of LSTM further strengthens the methodology by addressing the complexity of temporal dependencies, leveraging gating mechanisms to retain long-term information while alleviating the vanishing gradient problem. Moreover, the dimensionality transformations introduced throughout the process ensure compatibility and efficient feature extraction, reflecting careful consideration of the data's structure. Collectively, these elements provide a comprehensive and thoughtful approach to addressing the challenges of spatiotemporal data modeling.

The methodology is clearly structured and demonstrates an understanding of spatiotemporal data modeling. However, the description could benefit from deeper justifications, comparisons with state-of-the-art methods, and discussions on practical applicability and computational efficiency. Including these elements would make the section more robust and impactful.

Points for improvement

A. Clarity and Depth

The explanation includes detailed equations, but their connection to the overall architecture and contribution to the final prediction could be elaborated more clearly. For instance:

How does each module contribute quantitatively to improving prediction accuracy (e.g., ablation studies or comparisons)?

Is there evidence that dimensionality transformations improve performance compared to standard input formats?

B. Novelty

While the methodology combines known components (MSCNN, CBAM, and LSTM), the novelty appears to hinge on using multi-scale convolutions in both MSCNN and MSCBAM.

Could the author compare this novelty to existing approaches, particularly hybrid models combining CNNs, attention mechanisms, and LSTMs? For example, how does this model outperform others in literature, such as transformer-based models for spatiotemporal data?

C. Justification of Design Choices

The rationale for choosing specific kernel sizes (1×1, 3×3, 5×5) and their effect on performance isn't substantiated by experimental results in this section.

Why are these kernel sizes optimal for capturing multi-scale features? Could different combinations yield better results?

While the author claims that MSCBAM enhances interpretability, no specific discussion or metric is provided to demonstrate this interpretability or explain its practical significance.

D. Completeness

There is no mention of hyperparameter optimization, training details, or computational costs. These aspects are critical for reproducing the results and assessing the method's feasibility.

The method is described as generalizable, but it would be valuable to discuss its adaptability to datasets or problems beyond traffic flow prediction.

E. Connection to Objectives

The stated goal is to improve prediction accuracy, robustness, and nonlinear modeling capability. However, the methodology section doesn’t sufficiently address robustness or compare the nonlinear modeling capabilities of the proposed approach to existing methods.

F. Missing Comparisons with other decomposition

The methodology lacks references to alternative hybrid models and why MSCALSTM is preferable (see previous references in the point literature). For example:

Models combining wavelet transforms and neural networks for temporal feature extraction.

Transformer-based architectures for spatiotemporal data.

To improve the quality of the paper, I suggest the following points :

Add Justification for Key Design Choices: Provide theoretical or experimental justification for the choice of kernel sizes, attention mechanisms, and dimensionality transformations.

Enhance Clarity with Visuals: Include a detailed figure for MSCALSTM's overall structure, with annotations that highlight the interaction between MSCNN, MSCBAM, and LSTM.

Incorporate Quantitative Comparisons: Discuss ablation studies or comparative results showing the improvement brought by multi-scale convolutions and MSCBAM.

Expand on Robustness: Explain how the model handles noise, missing data, or varying input lengths, which are common challenges in time series prediction.

Address Computational Costs: Mention training times, hardware requirements, or the scalability of the approach for large datasets.

Experimental Analysis

Absence of Time Series Visualization: The lack of figures illustrating time series predictions (e.g., comparing actual vs. predicted values across models) diminishes the interpretability and tangible validation of the results. Graphical illustrations would offer a clearer understanding of model performance in capturing patterns and anomalies.

Unclear Result Section: The results are spread across subsections and lack a dedicated "Results" section, making it harder for readers to locate and interpret key findings succinctly. A focused subsection summarizing the comparative performance (e.g., with key takeaways from Tables 1 and 2) would enhance readability.

Impact of Computational Complexity: While MSCALSTM’s design is innovative, the computational burden introduced by multi-scale CNNs, CBAM, and attention mechanisms isn't thoroughly analyzed. A brief discussion on scalability for larger datasets would strengthen the evaluation.

Limited Baseline Model Interpretability: While baseline models like MAGRN and AGCRN are described briefly, their key advantages or limitations in relation to MSCALSTM aren't critically compared beyond performance metrics in Table 1.

Clarity in Ablation Study Results: The results of the ablation study (Table 2) are described but lack a detailed quantitative analysis in the text, making it harder to appreciate the precise impact of individual modules.

6. PLOS authors have the option to publish the peer review history of their article (what does this mean?). If published, this will include your full peer review and any attached files.

Reviewer #1: **Yes: **JC Li lijiangch@163.com

Reviewer #2: No

---

## [Author Response · Author response to Decision Letter 1]

7 Feb 2025

Dear Editor and Reviewers, I have revised the manuscript as requested. The specific changes are detailed in the “Response to Reviewers”.

---

## [Decision Letter · Decision Letter 1]

11 Mar 2025

PONE-D-24-51425R1Multi-scale time series prediction model based on deep learning and its applicationPLOS ONE

Dear Dr. Yang,

Thank you for submitting your manuscript to PLOS ONE. After careful consideration, we feel that it has merit but does not fully meet PLOS ONE’s publication criteria as it currently stands. Therefore, we invite you to submit a revised version of the manuscript that addresses the points raised during the review process.

**MINOR REVISION:**
Some minor improvements are still needed. Please address the comments by the reviewers. Thank you.

We look forward to receiving your revised manuscript.

Kind regards,

Zeyar Aung

Academic Editor

PLOS ONE

Journal Requirements:

Reviewers' comments:

Reviewer's Responses to Questions

**Comments to the Author**

1. If the authors have adequately addressed your comments raised in a previous round of review and you feel that this manuscript is now acceptable for publication, you may indicate that here to bypass the “Comments to the Author” section, enter your conflict of interest statement in the “Confidential to Editor” section, and submit your "Accept" recommendation.

Reviewer #1: All comments have been addressed

Reviewer #3: (No Response)

Reviewer #4: (No Response)

2. Is the manuscript technically sound, and do the data support the conclusions?

Reviewer #1: Partly

Reviewer #3: Yes

Reviewer #4: (No Response)

3. Has the statistical analysis been performed appropriately and rigorously? 

Reviewer #1: (No Response)

Reviewer #3: Yes

Reviewer #4: (No Response)

4. Have the authors made all data underlying the findings in their manuscript fully available?

Reviewer #1: (No Response)

Reviewer #3: Yes

Reviewer #4: (No Response)

5. Is the manuscript presented in an intelligible fashion and written in standard English?

Reviewer #1: (No Response)

Reviewer #3: Yes

Reviewer #4: (No Response)

6. Review Comments to the Author

Reviewer #1: Report

Manuscript Number: PONE-D-24-51425R1

Title: Multi-scale time series prediction model based on deep learning and its application

The paper proposes a multi-scale time series prediction model based on deep learning(MSCALSTM),aiming to enhance the accuracy and robustness of traffic flow forecasting.The authors have thoroughly addressed the previous review comments in the revised version,providing comprehensive supplements and optimizations to the model structure,experimental design,and results analysis.Specifically,in the integration of the multi-scale convolutional neural network(MSCNN),multi-scale convolutional block attention module(MSCBAM),and long short-term memory network(LSTM),the authors have validated the contributions of each module to model performance through detailed experiments and demonstrated the model's superiority on datasets in traffic flow forecasting and the energy domain.Additionally,the authors have supplemented the selection of model hyperparameters,training details,and computational cost analysis,enhancing the completeness and reproducibility of the study.By adding new dataset experiments,incorporating AIC and BIC evaluation metrics,optimizing figure presentations,providing detailed experimental settings and training process descriptions,and elaborating on each module of the model,the overall quality of the paper has been significantly improved.However,despite these efforts,there are still some areas where the paper can be further enhanced.

1. Although the authors have supplemented the description of model training time and hardware requirements,the analysis of the model's scalability and computational efficiency on large-scale datasets remains insufficient.In practical applications,tasks such as traffic flow forecasting and energy forecasting often involve massive amounts of data,where computational efficiency and memory usage are critical factors.It is recommended that the authors further analyze the model's performance on datasets of varying sizes and explore potential optimization strategies,such as distributed computing,model compression,or sparsification,to enhance the model's practicality and scalability.

2. The visualization and interpretability of the attention mechanism are inadequate.Despite further discussion on the internal workings of the multi-scale convolutional block attention module(MSCBAM),the authors lack visualization and interpretability analysis of the attention mechanism.A key advantage of attention mechanisms is their ability to reveal the model's focus on input data,thereby enhancing model interpretability.It is suggested that the authors visualize attention weights or attention maps to illustrate the model's focus areas on real data,helping readers better understand how the model identifies and utilizes key features.

3. The authors have provided detailed experimental results in the revised version,but lack statistical significance analysis of the results.When comparing the performance of different models,relying solely on average error metrics(such as MAE and RMSE)may not be sufficient to demonstrate the model's superiority.It is recommended that the authors conduct statistical significance tests(such as t-tests or Wilcoxon tests)to verify whether the improvement in model performance is statistically significant.This will enhance the credibility and persuasiveness of the research findings.

Reviewer #3: The article has been revised and improved according to the suggestions of the reviewers, and has made significant progress. It is recommended to accept the article.

Reviewer #4: This article proposes a new time series prediction model with an adaptive mechanism and experimentally verifies that it improves the accuracy and robustness of prediction forecasts. The article flows smoothly and is logically clear. However, the following issues need to be addressed:

1. There are some grammatical errors and sentence structure problems, such as inconsistent punctuation of juxtapositions and loose logical connections in long sentences. It is recommended that the whole text be carefully checked for grammar and language correction to ensure that it is clear and in line with academic writing standards.

2. In the introduction section, the authors mention the limited ability of existing graph convolution methods to capture dynamic spatial relationships, but do not address recent innovative advances in spatio-temporal dependency modelling with graph neural networks. For example, Advanced Engineering Informatics (10.1016/j.aei.2025.103123) proposes an adaptive spatio-temporal modelling framework based on dynamic graph structures, whose multi-scale dynamic pattern extraction can further support the rationality of model design. In addition, the paper emphasizes the role of multi-scale convolution for feature extraction, but no comparative analysis with current cutting-edge methods in the field of feature extraction (e.g., Measurement, 10.1016/j.measurement.2024.115417) is presented, and it is suggested to add relevant discussions to highlight the innovative boundaries of this work.

7. PLOS authors have the option to publish the peer review history of their article (what does this mean?). If published, this will include your full peer review and any attached files.

Reviewer #1: No

Reviewer #3: No

Reviewer #4: No

---

## [Author Response · Author response to Decision Letter 2]

21 Apr 2025

Dear Reviewers,

I hope this message finds you well. I have revised the manuscript according to your valuable feedback and have addressed all the comments you raised. The revised manuscript has been uploaded as an attachment.

---

## [Decision Letter · Decision Letter 2]

14 May 2025

Multi-scale time series prediction model based on deep learning and its application

PONE-D-24-51425R2

Dear Dr. Yang,

We’re pleased to inform you that your manuscript has been judged scientifically suitable for publication and will be formally accepted for publication once it meets all outstanding technical requirements.

Kind regards,

Zeyar Aung

Academic Editor

PLOS ONE

Additional Editor Comments (optional):

Reviewers' comments:

Reviewer's Responses to Questions

**Comments to the Author**

1. If the authors have adequately addressed your comments raised in a previous round of review and you feel that this manuscript is now acceptable for publication, you may indicate that here to bypass the “Comments to the Author” section, enter your conflict of interest statement in the “Confidential to Editor” section, and submit your "Accept" recommendation.

Reviewer #1: All comments have been addressed

Reviewer #4: (No Response)

2. Is the manuscript technically sound, and do the data support the conclusions?

Reviewer #1: Partly

Reviewer #4: (No Response)

3. Has the statistical analysis been performed appropriately and rigorously? 

Reviewer #1: Yes

Reviewer #4: (No Response)

4. Have the authors made all data underlying the findings in their manuscript fully available?

Reviewer #1: Yes

Reviewer #4: (No Response)

5. Is the manuscript presented in an intelligible fashion and written in standard English?

Reviewer #1: Yes

Reviewer #4: (No Response)

6. Review Comments to the Author

Reviewer #1: In this revised manuscript, the authors have addressed all my questions, and I find this version acceptable.

Reviewer #4: The author has carefully revised the manuscript based on the reviewers' comments and has addressed my concerns. The quality of the manuscript has been improved, and it can now be published in its revised version.

7. PLOS authors have the option to publish the peer review history of their article (what does this mean?). If published, this will include your full peer review and any attached files.

Reviewer #1: **Yes: **Jiangcheng Li

Reviewer #4: No

---

## [Editor Report · Acceptance letter]

PONE-D-24-51425R2

PLOS ONE

Dear Dr. Yang,

I'm pleased to inform you that your manuscript has been deemed suitable for publication in PLOS ONE. Congratulations! Your manuscript is now being handed over to our production team.

Kind regards,

on behalf of

Dr. Zeyar Aung

Academic Editor

PLOS ONE